# Exploring environmental and climate features associated with yellow fever across space and time in the Brazilian Atlantic Forest biome

**Maíra G. Kersul**[1]*, **Filipe V. S. Abreu**[2], **Adriano Pinter**[3], **Fabrício S. Campos**[4,5], **Miguel de S. Andrade**[6], **Danilo S. Teixeira**[7], **Marco A. B. de Almeida**[8], **Paulo M. Roehe**[5], **Ana Claudia Franco**[5], **Aline A. S. Campos**[9], **George R. Albuquerque**[7], **Bergmann M. Ribeiro**[10], **Anaiá da P. Sevá**[7]

**1** Programa de Pós-Graduação em Ciência Animal da Universidade Estadual de Santa Cruz (UESC), Ilhéus, Bahia, Brazil, **2** Laboratório de Comportamento de Insetos, Instituto Federal do Norte de Minas Gerais Campus Salinas, Salinas, Minas Gerais, Brazil, **3** Instituto Pasteur—SES—São Paulo, São Paulo, Brazil, **4** Laboratório de Bioinformática e Biotecnologia, Universidade Federal do Tocantins (UFTO), Palmas, Tocantins, Brazil, **5** Laboratório de Virologia, Departamento de Microbiologia Instituto de Ciências Básicas da Saúde, Universidade Federal do Rio Grande do Sul, Porto Alegre, Rio Grande do Sul, Brazil, **6** Setor de Biologia Molecular, Sabin Diagnóstico e Saúde, Brasília, Distrito Federal, Brazil, **7** Departamento de Ciências Agrárias e Ambientais (DCAA), UESC, Ilhéus, Bahia, Brazil, **8** Organização Pan-Americana da Saúde/Organização Mundial da Saúde, Brasília, Distrito Federal, Brazil, **9** Secretaria Estadual de Saúde do Rio Grande do Sul, Centro Estadual de Vigilância em Saúde, Porto Alegre, Rio Grande do Sul, Brazil, **10** Departamento de Biologia Celular, Universidade de Brasília, Brasília, Distrito Federal, Brazil

* mgkersul@uesc.br

**Data Availability Statement:** All relevant data are within the manuscript and its Supporting Information files.

## Abstract

The Atlantic Forest Biome (AFB) creates an ideal environment for the proliferation of vector mosquitoes, such as *Haemagogus* and *Sabethes* species, which transmit the Yellow Fever virus (YFV) to both human and non-human primates (NHP) (particularly *Alouatta* sp. and *Callithrix* sp.). From 2016 to 2020, 748 fatal cases of YF in humans and 1,763 in NHPs were reported in this biome, following several years free from the disease. This underscores the imminent risk posed by the YFV. In this study, we examined the spatiotemporal distribution patterns of YF cases in both NHPs and humans across the entire AFB during the outbreak period, using a generalized linear mixed regression model (GLMM) at the municipal level. Our analysis examined factors associated with the spread of YFV, including environmental characteristics, climate conditions, human vaccination coverage, and the presence of two additional YFV-affected NHP species. The occurrence of epizootics has been directly associated with natural forest formations and the presence of species within the *Callithrix* genus. Additionally, epizootics have been shown to be directly associated with human prevalence. Furthermore, human prevalence showed an inverse correlation with urban areas, temporary croplands, and savannah and grassland areas. Further analyses using Moran's Index to incorporate the neighborhoods of municipalities with cases in each studied host revealed additional variables, such as altitude, which showed a positive correlation. Additionally, the occurrence of the disease in both hosts exhibited a spatio-temporal distribution pattern. To effectively mitigate the spread of the virus, it is necessary to proactively expand vaccination coverage, refine NHP surveillance strategies, and enhance entomological surveillance in both natural and modified environments.

**Funding:** This work was funded by the National Council for Scientific and Technological Development (Conselho Nacional de Desenvolvimento Científico e Tecnológico – CNPq, call CNPq/MS-SCTIE-Decit No 22/2019, process number 443215/2019-7) and by the Coordination for the Improvement of Higher Education Personnel (CAPES), which is responsible for the fellowship awarded to Maíra G. Kersul (process number 88887.604207/2021-00). The funder had no role in this study design, data collection and analysis, decision to publish, or preparation of the manuscript.

**Competing interests:** We declare that we have no known competing personal relationships or financial interests that could appear to have influenced the work reported in this paper.

## Introduction

The Yellow Fever virus (YFV), belonging to the *Orthoflavivirus* genus [1], is a mosquito-borne pathogen which causes yellow fever (YF), a potentially fatal hemorrhagic fever. The disease has a lethality rate of approximately 20% in Africa's tropical regions and between 40–60% in South America [2, 3]. In Brazil, YF is endemic to the Amazon biome [3, 4]. The Atlantic Forest biome (AFB), another Brazilian tropical forest area that spans numerous states in Brazil and extends into parts of Argentina and Paraguay, was not reported to have YFV cases from 1940 until the decade from 2000 to 2010. However, multiple outbreaks occurred in or around the AFB in the years 2001, 2003, 2008, and 2009 [5–8]. In 2014, Brazil experienced an increase in the number of cases, and by 2016, the virus had spread, impacting susceptible populations in areas that were previously non-endemic within the AFB. This occurred in areas where the vaccine was either not recommended or failed to reach the entire population, resulting in inadequate vaccine coverage in regions that, despite having the highest human population density in the country, were left vulnerable [9–12]. Between 2016 and 2020, there were 748 confirmed human deaths and 1,763 non-human primate (NHP) deaths due to YFV [5]. This significant mortality rate triggered a global alert [5], raising concerns about the potential for international spread and the threat of re-urbanization of the virus [13–17]. After these events, the vaccination recommendation was extended to the entire country [18].

The re-emergence of YF beyond the Amazon biome unfolded in sporadic waves over an eight-year period, representing an unprecedented occurrence from 2014 to 2022. The peak incidence was observed between 2016 and 2020 in the AFB. During that period, thousands of humans and NHPs were impacted, making it the most significant YF outbreak in the last 80 years. This outbreak was especially concerning compared to others in certain AFB areas due to the high population density and the risk of re-urbanization [11, 16, 17, 19]. In addition to spreading to regions near major urban centers, YFV was already present in NHPs [20, 21] several months before the first human cases were reported. NHPs are crucial as sentinels in YFV surveillance programs for human cases [22–24].

The persistence of the virus in a specific environment depends on the availability of infected hosts, the dispersal capacity of vectors, and favorable habitat conditions [25]. Unlike NHPs, which cannot sustain acute infections over long distances [16], infected vectors have the capability to travel considerable distances, potentially covering up to 11.5 km [26, 27]. Surveillance of YF has demonstrated that the disease can move across territories and may cover up to 500 km in six months [7]. This assessment was further validated by viral genomic surveillance, which estimated that virus lineages could spread at an average rate of 0.5 to 4.25 kilometers per day [20, 28, 29].

In the Americas, YFV is associated with two distinct epidemiological cycles: sylvatic and urban [30]. In the sylvatic cycle, YFV spreads between NHPs and diurnal mosquitoes, mainly *Haemagogus* spp. and *Sabethes* spp., with human transmission occurring through accidental exposure [30, 31]. The urban transmission cycle, primarily driven by the *Aedes aegypti* mosquito as the primary vector and humans as vertebrate hosts, has not been reported in Brazil since 1942 [16]. The high densities of *Aedes aegypti* across Brazilian cities highlight the potential risk of YFV reintroduction into *Aedes*-infested areas, which could lead to future outbreaks and a catastrophic re-urbanization scenario in Brazil, especially if YFV outbreaks reach the vicinity of major urban areas [15, 16].

The spatiotemporal dynamics of YFV are primarily influenced by the interplay of physiological interactions between the virus and its vector, the virus and its vertebrate hosts, as well as the population dynamics of both the vector and the hosts [32]. Human activities are increasingly placing pressure on the environment and biodiversity [33–36], which can significantly

disrupt the complex interactions among hosts, vectors, and pathogens. This disruption facilitates the spread of mosquito-borne viruses such as YFV by creating conditions that increase the risk of their dispersion [32, 33, 37, 38]. Furthermore, climatic changes significantly influence the distribution and density of mosquitoes, thereby affecting vector density and impacting YFV transmission intensity [39, 40]. The occurrence of YF in 2017–2018 may have been influenced by stable air humidity, high temperatures, and significant rainfall [16, 31].

In summary, environmental factors (including land use and vegetation cover), climatic features, and human immunization strategy act as crucial variables in the interplay between the virus and hosts [25, 33, 41, 42]. Understanding the geographic distribution of the disease and associated variables is crucial for identifying the drivers of disease occurrence. These analyses can inform future strategies, investigations, and predictions regarding the spread of YFV, enabling a comprehensive assessment of mitigation and surveillance measures. The present study seeks to analyze the spatiotemporal patterns of YF cases in NHPs and humans during the 2016–2020 outbreak impacting the Atlantic Forest biome in Brazil. Additionally, we aim to identify and characterize risk factors and preventative measures that can inform public policies designed to limit the dispersal of YFV.

## Methods

### 1. Study description

The Brazilian municipalities within the Atlantic Forest biome (AFB) cover an area of approximately 1,384,940 km$^2$, extending along the Atlantic coast from the state of Rio Grande do Norte in the northeast to the state of Rio Grande do Sul in the south. Biomes are characterized by features such as vegetation, physical conditions, and biological diversity. Their boundaries were determined using a public domain 2019 shapefile from IBGE [43]. The biome of each municipality was determined by using both their shapefiles and the intersection function in QGIS. For a comprehensive analysis, we focused on cases of YF in non-human primates (NHPs) and humans from 2016 to 2020, a period during which we had the most consistent data for this biome. We used a spatio-temporal regression analysis incorporating a generalized linear mixed model and Moran's spatial correlation to evaluate potential factors linked to the incidence of YF cases. The characteristics of key factors associated with the disease were compared between municipalities that had confirmed cases of YF and those that did not.

### 2. Data collection

**2.1 YF in NHPs and humans.** The Brazilian federal health authorities report cases of YF in NHPs and humans per municipality, which are the smallest administrative units for implementing health policies [44, 45]. Data on YF occurrences in NHPs and the municipalities identified as probable places of infection (PPI) in human cases were sourced from the Brazilian Ministry of Health (BMH) website [5]. Additionally, information regarding the date of first symptoms, age, and sex of the human cases was also obtained from the same source. Human cases can be confirmed through laboratory tests or epidemiological links, in accordance with BMH guidelines [45, 46]. Epizootics in NHPs were confirmed either by a positive laboratory result for YFV in at least one infected animal at the PPI or through an epidemiological link. This link could be with another confirmed epizootic, a human case, or virus detection in mosquitoes, as recommended by the BMH [30, 45, 46]. In the context of this study, data on human and NHP cases were obtained from the routine surveillance conducted by the Health Departments of various municipalities and states, in accordance with the guidelines established by the BMH and the Brazilian National Committee for Ethics in Research [47]. The human data

do not contain personally identifiable information on the case report forms, thereby obviating the need for authorization by the Brazilian Ethics Committee [48].

To account for the wide variations in territorial sizes and population numbers across different municipalities, we analyzed the rate of cases per capita rather than their absolute numbers. The prevalence of YF in humans was determined by calculating the number of cases per 10,000 inhabitants in each municipality annually, based on the total population figures from the 2010 census conducted by the Brazilian Institute of Geography and Statistics (IBGE). This entire population was considered, rather than focusing solely on rural areas, because individuals can become infected in rural or wild areas and still be urban residents. The cases of NHP are characterized by the annual occurrence of confirmed YFV cases in each municipality and are referred to as epizootics. This variable is characterized by its presence or absence (S1 Table).

The dynamics of YFV are closely associated with environmental factors, primarily because the vector is sensitive to these conditions [11, 33, 42, 49–54]. Consequently, the municipalities selected for this study were chosen according to biome classifications [43], which reflect their climatic and environmental similarities. This approach was considered more reliable than depending solely on political divisions such as states or regions. Notably, among Brazil's six biomes, the AFB was the most significantly impacted by the outbreak during the period from 2016 to 2020, which our analysis covers. This was evident from the increased number of municipalities with confirmed YF cases in both humans and NHPs within the AFB (S2 Table). In the AFB, more than 10% of the 3,079 municipalities experienced the outbreak. The data show that 367 municipalities reported confirmed cases of YF in NHPs (n = 411, considering 40 municipalities counted twice due to recorded cases for more than one year) and 375 municipalities had confirmed cases in humans (n = 425, with 50 municipalities counted twice for the same reason).

**2.2 Associated factors.** The factors believed to influence the incidence of YF in NHPs and humans were investigated. These factors included: 1) environmental characteristics such as land use and vegetation cover, which encompass both reforestation and deforestation within forest formations and wooded Restinga areas; 2) climate factors; 3) the two predominant climate classifications in AFB according to the Köppen index; 4) human vaccination efforts; and 5) the presence of two primary NHP species impacted by YFV, specifically *Alouatta* sp. and *Callithrix* sp. (S1 Table).

Data on land use and vegetation cover were obtained from the public domain data of MapBiomas (Collection 4.0), which provides the area size for each category by municipality in a tabular format. The extraction of these values was performed using satellite imagery in raster format with a resolution of 30-meter pixels. Data manipulation was conducted using R software (version 3.6.1). This involved extracting annual values and calculating the proportion of area covered by each category within each municipality. To ensure an adequate timeframe for the establishment of contact between hosts and wild vectors, which could lead to new cases, the rate of deforestation was calculated annually. This calculation represented the percentage decrease in forested area from year $t$ to year $t$-1, focusing solely on values that were equal to or less than zero, indicating a reduction in area (Eq. 1).

$$Deforestation_t = 100 \times ((t - (t - 1)) \div (t - 1)) \qquad \text{(Eq1)}$$

In the analysis, both savannah and grassland vegetation types were considered. These are prominent in the Cerrado and Caatinga biomes, particularly in the ecotone regions with AFB [55]. It is essential to consider these transition zones, as they significantly influence the spread of YFV.

The Fragmentation Index for AFB forests was calculated using the shapefile of fragments, which was generated by polygonizing raster images from MapBiomas in QGIS software. This index was calculated using the Perimetral Index, which is defined as the ratio of perimeter to area (perimeter:area) for the total forest fragments in each municipality [56]. This index is used to assess the complexity of polygons (forest fragments), where the perimeter-to-area ratio inversely correlates with size [57].

Climate data, including temperature, rainfall, and humidity, were obtained from the Copernicus.eu dataset, which features a spatial raster resolution of 0.25˚ x 0.25˚ (~28 km$^2$). The monthly mean pixel values for each municipality were calculated using the zonal statistics function in QGIS software. Monthly values were averaged to derive annual figures, and the annual range for each variable was calculated as the difference between the maximum and minimum monthly values. Temperature was converted to degrees Celsius, and rainfall to millimeters of precipitation per day (mm/day).

Two common climate classification types, "Aw" and "Cfa," as defined by Köppen, were selected for each AFB municipality. These classifications take into account patterns associated with temperature, precipitation, altitude, and latitude. The term "Aw" denotes a tropical climate characterized by a dry winter, whereas "Cfa" represents a humid subtropical climate with hot summers. This data, provided in a raster format with a spatial resolution of 90 meters, was sourced from weather stations monitored by the Food and Agriculture Organization of the United Nations (FAO/ONU). The altitude values used in our study, independently sourced from the same author, were originally compiled from multiple sources and summarized by Alvares et al. [58].

For the vaccination features, we utilized Infant Vaccine Coverage (IVC), which is the recommended policy for child immunization across Brazil [59]. This data was sourced from the TABNET health website, managed by BMH [60]. To provide context, until July 2017, health authorities recommended administering the first vaccine dose to children at nine months of age, followed by a booster every 10 years. Subsequently, because the vaccine provides long-lasting immunity, it is recommended that children receive the first dose at nine months and a booster at four years of age. Children aged five and older should receive a single dose. Before 2017, 63.3% (3529 out of 5571) of the Brazilian municipalities adhered to these vaccine recommendations, increasing to 80.2% (4469 out of 5571) in the same year [61], and this trend persisted until 2020 (see S1 Fig).

The occurrence data for NHP species were sourced from a dataset provided by Culot et al. [62]. The occurrence of each species in each municipality was recorded as a binary value.

## 3. Descriptive analysis

**3.1. YF cases and municipalities.** *i*) presented the total number of YF cases in NHPs and humans; *ii*) examined the scope of municipalities impacted by YF; and *iii*) performed a comparative analysis of these data across different study years.

**3.2. YF human cases behavior.** *i*) used a sex and age pyramid to describe the behavior of YF human cases and *ii*) implemented the R program with the epiDisplay package, specifically using the pyramid function.

**3.3. YF spread.** *i*) investigated the spread of epizootics and the prevalence of YFV in humans across different regions and over time; *ii*) examined the longest durations of YF cases to identify patterns and trends.

**3.4. Distribution of YF cases among AFB municipalities by state.** *i*) ranked states according to the number of YF cases; *ii*) assessed the ranking of states based on the number of municipalities affected by YF; and *iii*) calculated the human prevalence rate (cases per 10,000 inhabitants) in municipalities with confirmed cases within each state.

**3.5. Persistence or recurrence of municipalities.** *i*) investigated the persistence or recurrence of municipalities reporting YF cases; *ii*) analyzed municipalities that reported cases for more than one consecutive year; *iii*) identified municipalities where both NHP and human YF cases were recorded over two consecutive years, particularly those persisting for over six months; and *iv*) examined the timing of human cases, correlating these observations with vaccination efforts. This approach aligns with the findings of Romano et al. [24], which indicate that human YF cases predominantly occur from December to March in areas with AFB.

## 4. Modeling/Statistical analysis

To evaluate the relationship between epizootics and human prevalence (dependent variables) and various factors such as environmental conditions, climate, infant vaccination coverage, and the presence of NHP species (independent variables), we used a spatio-temporal generalized linear mixed model (GLMM) for each variable.

**4.1 Variables of the model.** For each model, we considered the presence of epizootics and human prevalence as dependent variables. The term "epizootics presence" refers to municipalities where YF cases have been confirmed in NHPs.

Since the dependent variable, human prevalence, did not exhibit a normal distribution, as evidenced by the QQ plot (S2A Fig), we applied a log transformation (log (1 + dependent variable)) to normalize the skewed data and reduce the substantial variability across municipalities (S2B Fig) [63]. Given the spatial distribution of the disease, marked by a high number of municipalities with no cases and a few with numerous cases, it was crucial to employ models from the Poisson family. This choice is consistent with the exponential distribution commonly associated with this type of spatial pattern.

Among all the independent variables considered for this study, a rigorous selection process was undertaken to determine which variables would be included in each model and to address any residual accuracy errors. This selection process comprised three sequential analyses: 1) Biological Plausibility: We assessed the biological plausibility of each variable having a causal relationship with the respective dependent variable, whether in NHP) or humans; 2) Mapping: We mapped the distribution of each variable across the municipalities within the AFB to visualize variations in their distribution. Variables with uncommon or less obvious distributions were not included. Mapping activities were conducted using QGIS software (version 3.18); 3) Correlation analysis: The correlation among variables was assessed using the Spearman ($\rho$) test due to their non-normal distribution. Independent variables that demonstrated a significant correlation ($p < 0.05$) with the dependent variable were selected. Among these, variables that were strongly correlated with each other ($\rho > 0.20$ or $\rho < -0.20$ and $p < 0.05$) were further scrutinized, and the variable with the highest significance to the dependent variable was chosen. After defining these criteria, variables were sorted based on their strong correlation with the dependent variable and sequentially included in the model using a backward stepwise selection process.

In the initial phase, which concentrated on biological plausibility, the following potential variables were identified for inclusion in both models: 1) Average temperature; 2) Temperature range; 3) Average rainfall; 4) Rainfall range; 5) Average humidity; 6) Humidity range; 7) Altitude; 8) Presence of Köppen climate classification "Aw"; 9) Presence of Köppen climate classification "Cfa"; 10) Agropastoral land use: agriculture and pasture; 11) Forest formations; 12) Wooded restinga; 13) Deforestation in forest formations; 14) Deforestation in wooded restinga; 15) Other non-forest formations; 16) Perennial crops; 17) Temporary crops; 18) Savannahs and grasslands; 19) Urban areas: urban and other non-vegetated areas; 20) Water bodies: wetlands, rivers, lakes, and oceans; 21) Forest fragmentation index; 22) Infant vaccine

coverage; 23) Occurrence of *Alouatta* sp.; 24) Occurrence of *Callithrix* sp.; 25) For the analysis of human prevalence, the term "epizootics" was included to account for the observation that human cases typically follow outbreaks in NHPs [20, 22–24]. See S2 Table for more details.

**4.2 Generalized linear mixed model.** The GLMMs for each host included fixed effects, which comprised environmental, social, and climatic variables, as well as random effects that accounted for temporal (year) and spatial (municipalities) variations. This model was adapted from [64–66] and is expressed by the following equation (Eq. 2):

$$Log[E(Y_{it})] = \alpha_i + \beta X_{it} + \alpha^{d_{ij}} \tag{Eq2}$$

Where $Y_{it}$ is the log-transformed disease prevalence in municipality $i$ during year $t$ (where $i$ = 1, 2, 3, . . ., $n$ and $t$ = 2016, 2017, . . ., 2020), $a_i$ represents the random intercept for municipality $i$ (random effect of space); and $\beta X_{it}$ denotes the fixed effects (independent variables) in municipality $i$ during year $t$. To control residual spatial correlation, the $\alpha$ parameter of $d_{ij}$ was calculated to represent the inverse distance between the centroids of municipalities $i$ and their neighboring municipalities $j$. This spatial correlation structure assumes that the correlation of variables between municipalities decreases exponentially as spatial distance increases.

The models were implemented in R software (version 3.6.1), using the MASS package and the glmmPQL function. The code applicable to both NHPs and humans is presented below. The complete script for variable analysis and model development is available in the S1 Appendix.

Model of NHP = glmmPQL(epizootics) ~ generic independent variables,
data = YellowFever_ AtlanticForest_table,
random = ~1|year|municipality_ID,
corr = corSpatial (form = ~jitter(longitude) + latitude, type = "exponential"),
family = binomial)
Model of Human = glmmPQL(log(1 + human prevalence) ~ generic independent variables,
data = YellowFever_ AtlanticForest_table,
random = ~1|year|municipality_ID,
corr = corSpatial (form = ~jitter(longitude) + latitude, type = "exponential"),
family = poisson)

After running the models and identifying significant variables for each host (NHP and human), we investigated the values of these variables in municipalities with confirmed cases (MCC) and those without confirmed cases (MNC). This analysis sought to identify and characterize the behavioral patterns of these variables across different categories. Specifically, we analyzed their measures of central tendency to gain insights into the distribution and characteristics of these variables in the context of confirmed YF cases.

## 5. Global Moran's spatial analysis

To explore the spatial distribution patterns of YF epizootics and the log-transformed prevalence in humans separately, we performed both Global and Local Moran's I univariate analyses to examine spatial autocorrelation. This analysis enabled us to determine the similarity of YF cases in one municipality to those in the surrounding areas. Additionally, we conducted both global and local Moran's bivariate analyses to assess the correlations between YF hosts and YF variables (such as epizootics or human prevalence) and independent variables. This bivariate analysis examined the impact of the YF host variable in each municipality and its influence from neighboring municipalities.

For all variables, we calculated the average values spanning the years 2016 to 2020. The neighborhood matrix was constructed by defining the elements as the inverse of the distance

between the centroids of the municipalities, including all those within a radius of 136.88 km. This radius was chosen to ensure that each municipality had at least one neighboring municipality. The assumed spatial correlation structure posits that the power of correlation diminishes with increasing spatial distance. In all analyses, we assessed the significance of the results using randomization with 999 permutations, considering a p-value of less than 0.05 as statistically significant. The software GeoDa © v. 1.14.0 was used for these spatial analyses.

For the analysis of epizootics, unlike in GLMM, we treated the urban area as the independent variable. This approach aimed to assess the impact of urban areas in neighboring municipalities (*j*) on epizootic events in a specific target municipality *i*.

## Results

### 1. YF cases in NHPs and humans

Over the entire study period, 93.23% (1,763/1,891) of all reported YF cases in NHPs occurred in the AFB. The highest number of cases within this biome was recorded in 2017, reaching 45.66% (805) cases. Similarly, out of all reported Brazilian human YF cases, 98.10% (2,224/2,267) occurred in the AFB. In 2018, the highest prevalence was recorded at 58.63%, with 1,304 reported cases, corresponding to 0.098 cases per 10,000 humans.

In the AFB, males accounted for 82.96% of human YF cases, totaling 1,845 instances. Most of affected individuals were adults, comprising 74.04% of the cases (1,574 cases), predominantly within the 20–59 year age group. Elderly individuals (age >60 years), represented 19.57% of the cases (416 cases), followed by adolescents (4.52%; 96 cases; age 13–19 years), children (1.74%; 37 cases; age 1–12 years) and infants (0.14%; 3 cases; age <1 year). Approximately 4.41% of cases (98 in total) had missing or incomplete information regarding sex and age. Notably, male adults were the group most affected by YFV during the study period (Fig 1).

### 2. YF cases in AFB municipalities

During the study period, YF cases (NHP, human, or both) were reported in 18.77% (578 out of 3,079) of the municipalities within the AFB. Fig 2 illustrates a significant increase in the number of municipalities affected in 2017 and 2018, particularly highlighting those with concurrent NHP and human cases. This peak in concurrent occurrences coincided with the year that had the highest number of affected municipalities. The incidence of YF in NHPs increased in 2017, but subsequently decreased from 2018 to 2019. In 2020, there was a resurgence, coinciding with an expanded effort to surveil and detect epizootics [9]. In contrast, municipalities that reported only human cases of YF saw an increase in 2017 and 2018, followed by a subsequent decline in 2019 and 2020.

Spatiotemporal maps (Fig 3 for epizootics and Fig 4 for human prevalence) vividly illustrate the rapid spread of YF in both animal and human populations within the AFB, progressing from the central to the southern regions. In terms of human prevalence, 21 municipalities fell into higher categories, each reporting over 109 cases per 10,000 inhabitants. The areas with high prevalence were mainly located in the states of São Paulo (SP), Espírito Santo (ES), and Minas Gerais (MG).

Considering the timeline of the spread across states (Fig 5), it is clear that Espírito Santo and Rio de Janeiro were particularly vulnerable to YFV in 2017 and 2018, respectively. These states had a higher proportion of municipalities with both NHP and human cases, suggesting significant interaction between epizootic cycles and unvaccinated human populations. In 2020, there was a significant reduction in the percentage of municipalities reporting human cases of YF in the three states of Paraná, Santa Catarina, and São Paulo.

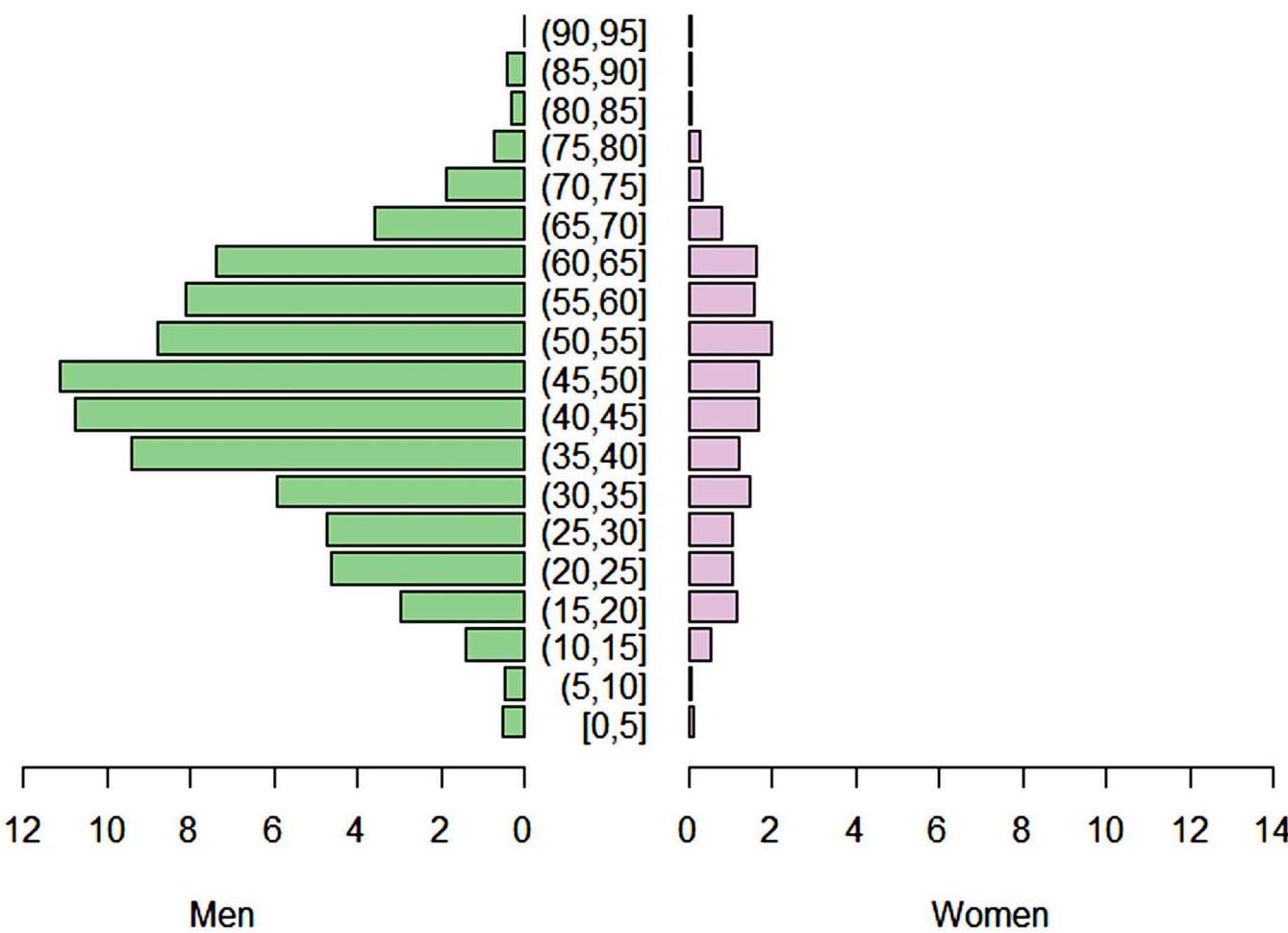

**Fig 1. Age pyramid of YF human cases by sex between 2016 and 2020 in AFB.**

When the analysis was expanded to include all 1,763 YF cases in NHP across municipalities within the AFB that had confirmed cases (367), the state of São Paulo had the highest number of cases (48.21%), followed by Paraná (22.40%) and Minas Gerais (10.66%). Conversely, when considering all YF cases in humans (2,224), the majority were reported in Minas Gerais (43.79%), followed by São Paulo (29.41%) (Fig 6).

Over the course of the study period, and taking into account the number of municipalities in the AFB, Rio de Janeiro recorded the highest percentage of NHP cases (35.87%) and human cases (48.91%). Similarly, Espírito Santo showed high percentages of NHP cases (30.77%) and human cases (27.91%). When considering human prevalence, Minas Gerais had the highest rate among the confirmed municipalities (1.17 per 10,000 inhabitants), followed by Espírito Santo (0.95 per 10,000 inhabitants) (Table 1).

Municipalities that have reported confirmed cases of yellow fever (YF) in non-human primates (NHPs) and humans for over a year constitute 10.90% (40 out of 367) and 13.33% (50 out of 375) of the total, respectively (see S4 Table).

Notably, three municipalities demonstrated persistent occurrences spanning more than two years for NHP YF cases: Belo Horizonte (MG) in 2016, 2017, and 2018; Atibaia (SP) in 2017, 2018, and 2019; and São José do Rio Preto (SP) in 2016, 2017, 2018 and 2020. The municipalities that reported cases twice over an interval exceeding six months, spanning two

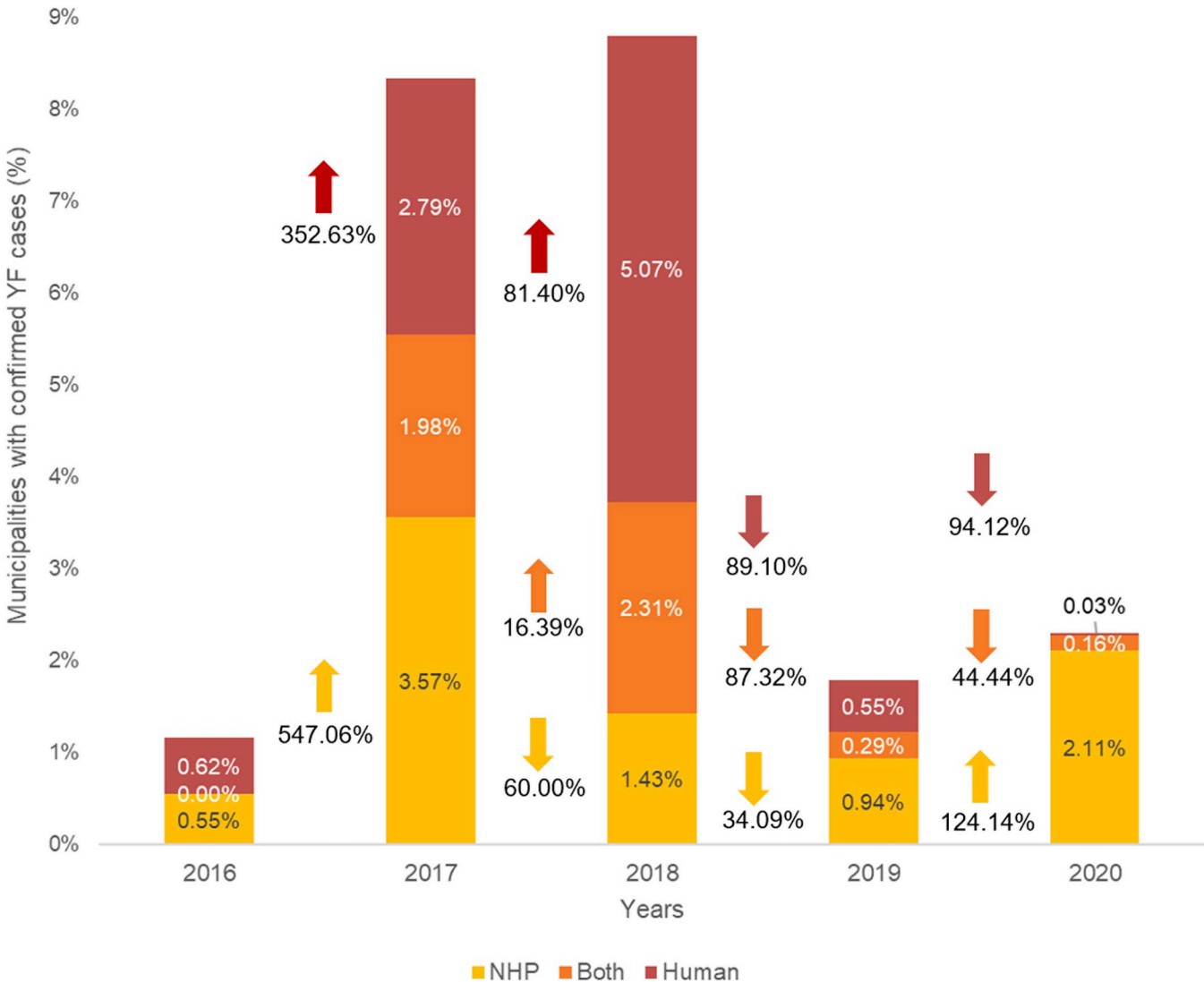

**Fig 2. Percentage of municipalities with confirmed cases of YF in NHPs, humans, and both combined over the years 2016 to 2020.** ↑: Indicates an increase in values from one year to the next.; ↓: Indicates a decrease in values from one year to the next.

consecutive years, represented 50% (20/40) of NHP cases and 32% (16/50) of human cases (see Figs 7 and 8, respectively).

In the regression model for NHPs, we observed a positive and significant association between the presence of *Callithrix* sp. and forest formation area (p < 0.05) (Table 2). In S3A and S3B Fig, we present the distribution of these specific variables.

In the model that predicts the prevalence of YF in humans, with human prevalence as the dependent variable, positive associations were observed with epizootics (see Fig 9 and S3C Fig). Conversely, negative associations were observed with temporary crops, savannahs and grasslands, and urban areas (see Table 2 and S3D–S3F Fig).

The distribution patterns of significant variables differed between municipalities with confirmed cases (MCC) and those without confirmed cases (MNC). In instances where there was a direct association with the dependent variables (epizootics and human prevalence), the highest values of the independent variables were observed in the MCC. Conversely, when the

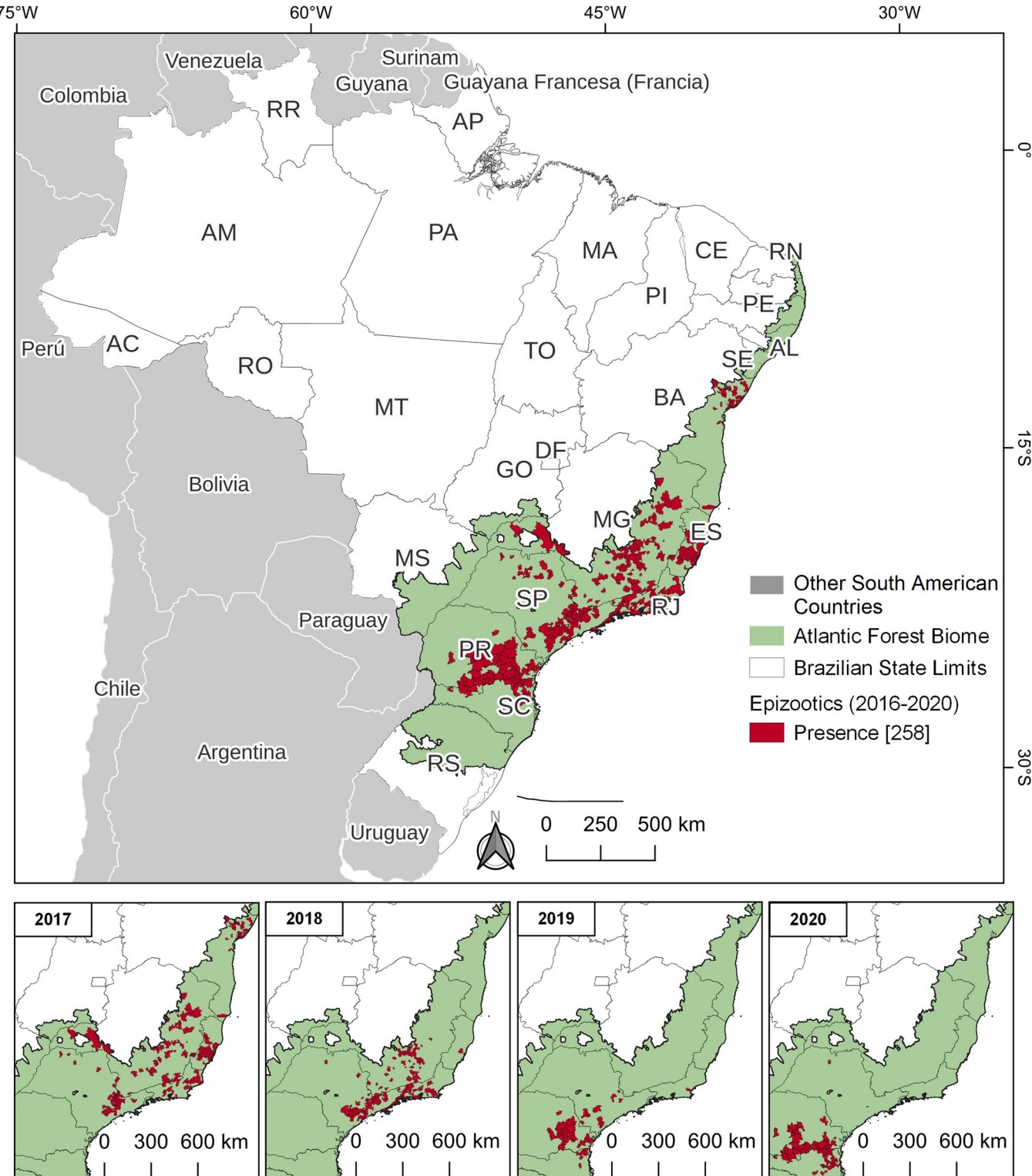

**Fig 3. Presence of epizootics in AFB from 2016 to 2020.** The count of municipalities within each category is indicated in parentheses. The annual demonstration omits 2016 due to the low number of cases, which did not significantly contribute to the analysis. The temporal maps begin in 2017, coinciding with a significant peak in NHP cases.

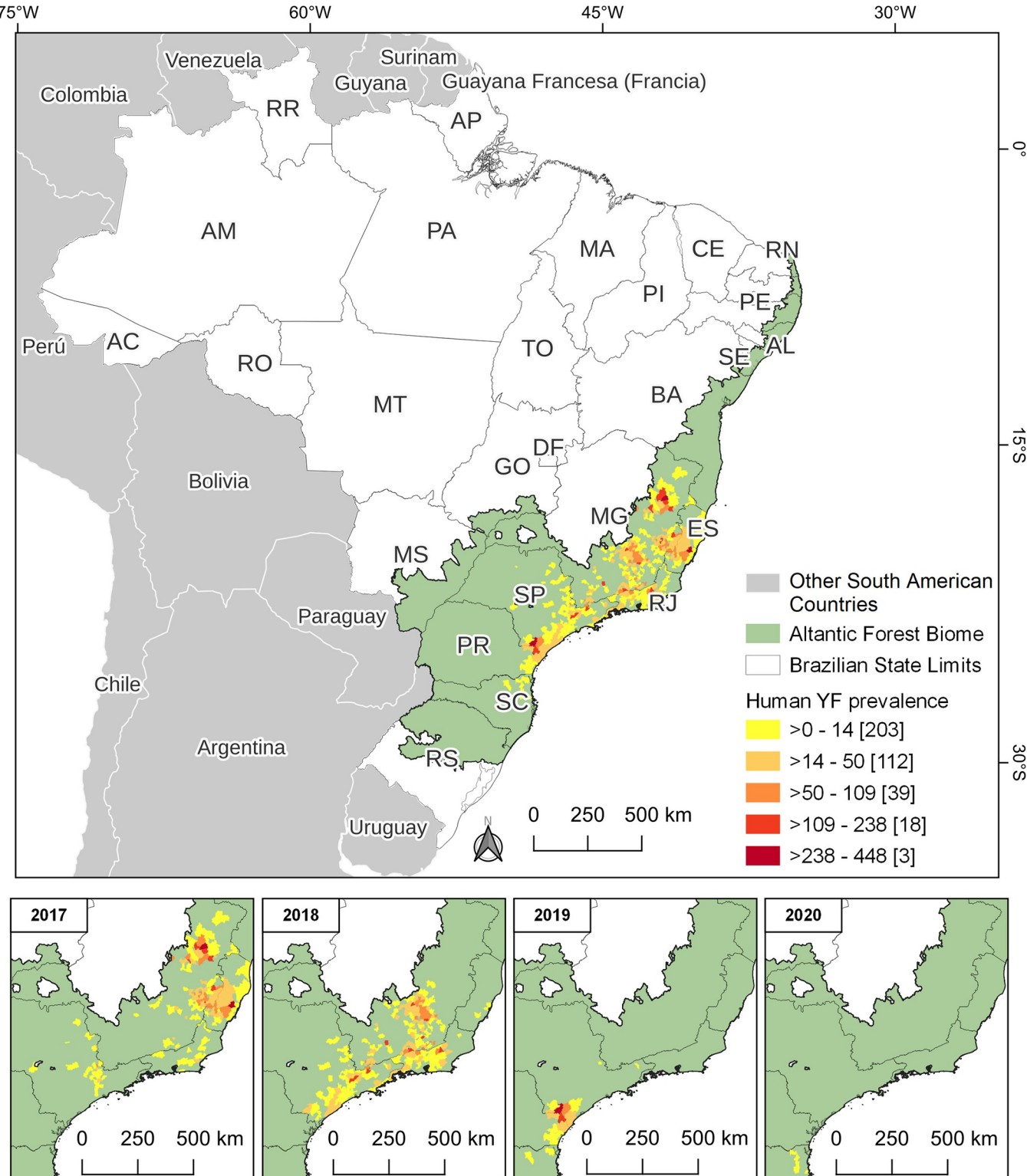

**Fig 4. Prevalence of YF in humans (cases per 10,000 inhabitants) in AFB from 2016 to 2020, categorized into five natural break classes.** The count of municipalities within each category is indicated in parentheses. The annual demonstration omits 2016 due to the low number of cases, which did not significantly contribute to the analysis. The temporal maps begin in 2017, coinciding with a significant peak in human cases.

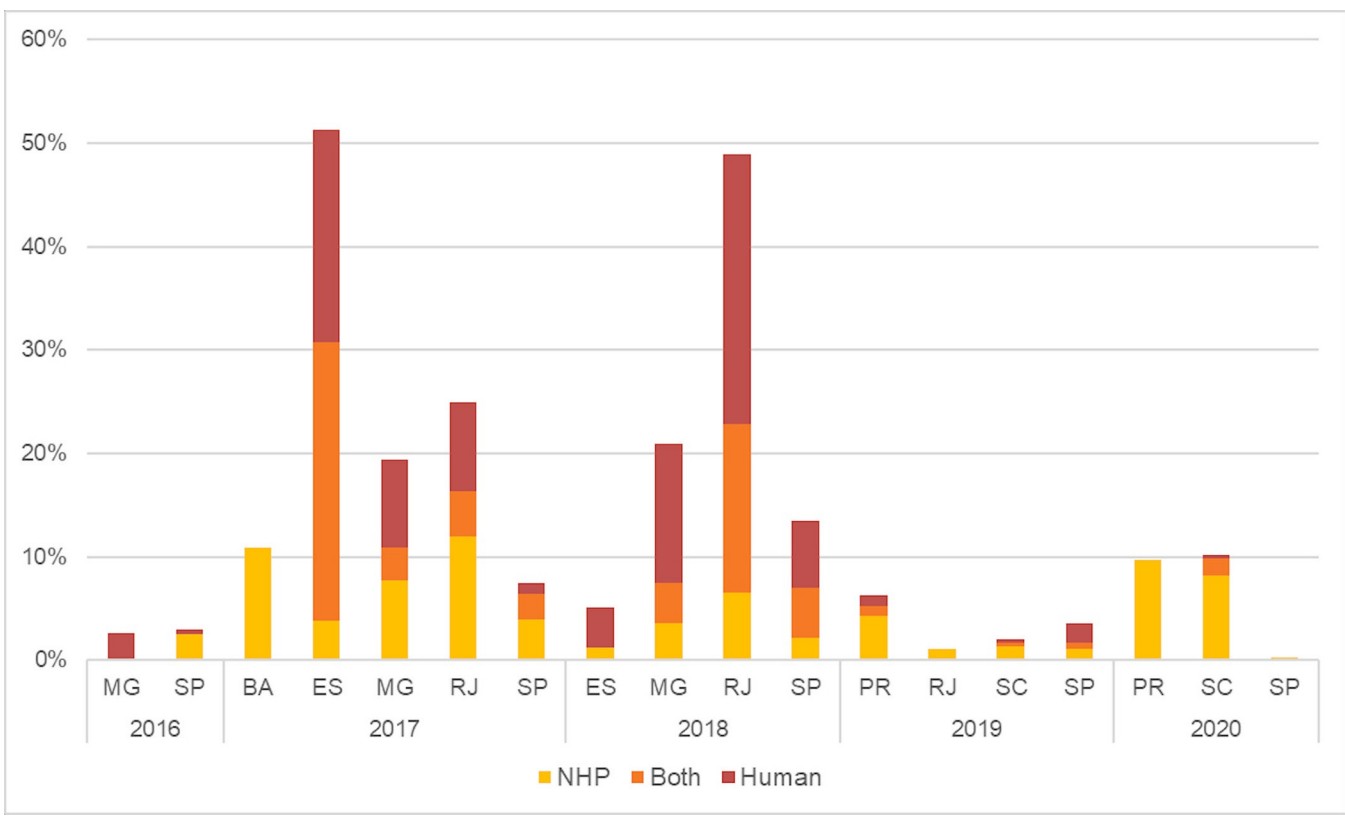

**Fig 5. Distribution of YF cases (%) in municipalities within the Brazilian states of the Amazon Forest biome.**

association was indirect, the highest values were observed in MNC (Table 3). In the urban area, however, the pattern differed; the relationship was inverse, with a higher percentage observed in MCC (average = 5.59%) compared to MNC (average = 3.2%).

The interactions among all independent variables that showed significant correlations ($p < 0.05$ and $\rho < -0.2$ or $> 0.2$) are detailed in S3 Table. Although some variables, such as the presence of *Alouatta* sp., temporary crops, and altitude in the NHP model, and IVC, presence of *Callithrix* sp., and forest formation in the Human model, did not exhibit a strong enough

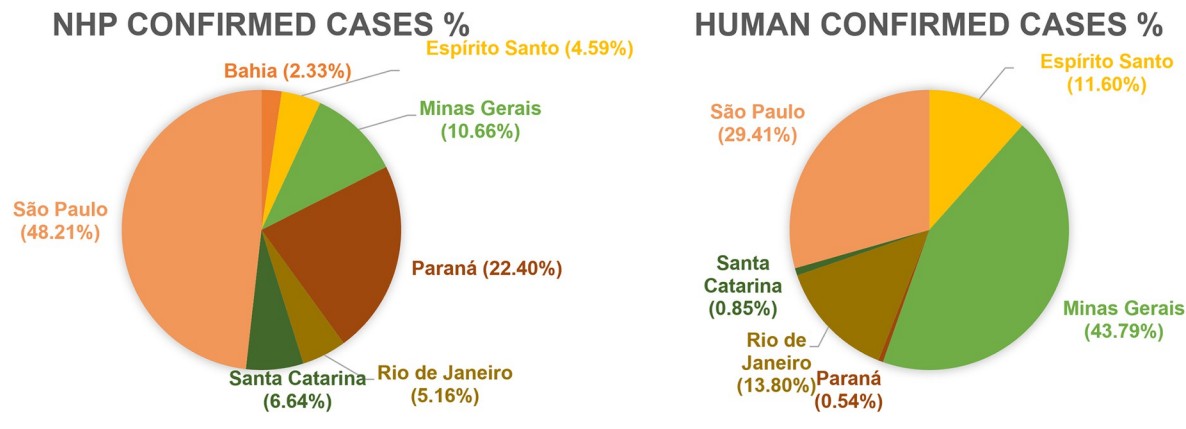

**Fig 6. Distribution of YF cases across Brazilian states from 2016 to 2020.**

**Table 1. Data by state: Municipalities in the AFB featuring NHPs alongside confirmed human cases in each state, and the prevalence of humans (cases/inhabitants).**

| State | Municipalities with confirmed NHP cases % (n/total) | Municipalities with confirmed Human cases % (n) | Human cases per 10,000 inhab |
|---|---|---|---|
| **Bahia (BA)** | 10.88% (21/193) | | |
| **Espírito Santo (ES)** | 30.77% (24/78) | 47.44% (37/78) | 0.95 (258) |
| **Minas Gerais (MG)** | 17.36% (112/645) | 27.91% (180/645) | 1.17 (974) |
| **Paraná (PR)** | 13.28% (53/399) | 2.01% (8/399) | 0.21 (12) |
| **Rio de Janeiro (RJ)** | 35.87% (33/92) | 48.91% (45/92) | 0.56 (307) |
| **Santa Catarina (SC)** | 10.51% (31/295) | 2.71% (8/295) | 0.14 (19) |
| **São Paulo (SP)** | 14.49% (93/642) | 15.11% (97/642) | 0.27 (654) |
| **Total** | **11.92% (367/3,079)** | **12.18% (375/3,079)** | **0.52 (2,224)** |

inhab: inhabitants

correlation with the dependent variable to be included in the model and were represented by other independent variables, they are still discussed due to the results of the Moran analysis and their biological relevance.

Using the univariate Global Moran's Index (GMI), we aimed to visualize the spatial dispersion of NHP and human cases. Both NHP and human cases formed clusters in the central region of the AFB. However, the distribution patterns differed; human cases were more concentrated than those of NHPs. Using bivariate spatial Moran analysis, we identified relevant variables with a GMI > 0.100 or <-0.100. Our findings indicate that epizootics are directly associated with factors such as altitude, forest formation area, urban area, and rainfall range. Conversely, an inverse association was observed with temporary crops, the Cfa Köppen index, and humidity range. In contrast, human cases showed a direct association with rainfall, altitude, and epizootics, and an inverse association with temporary crops, the Köppen Cfa index, and humidity levels (Table 4). The local features are illustrated in S4–S6 Figs. The term "inverse" suggests that higher values are typically associated with lower values and vice versa, indicating a pattern of greater dispersion.

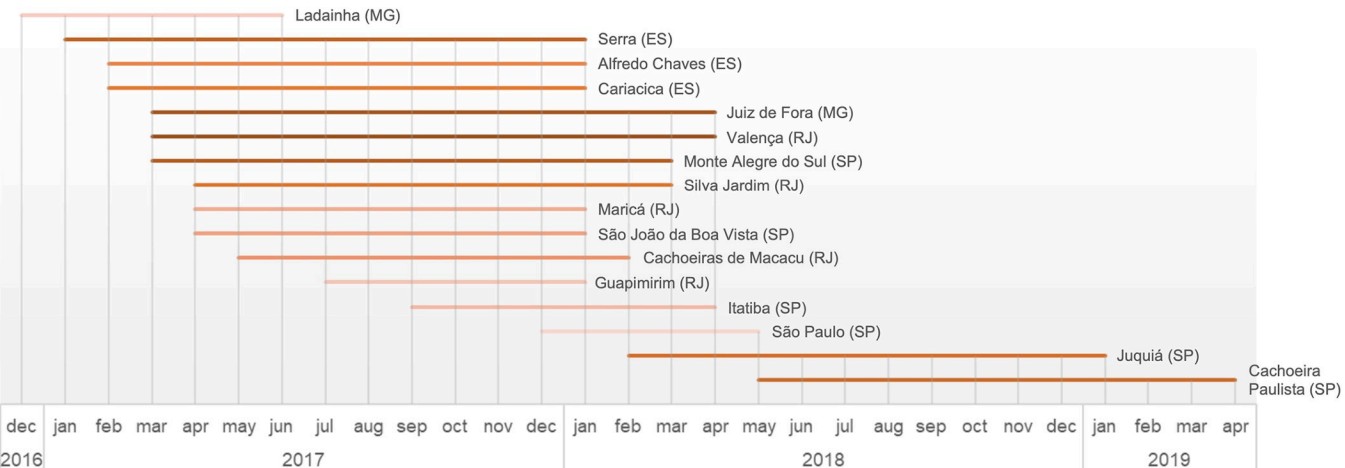

**Fig 7. Duration and distribution of NHP YF cases in municipalities.** The gradation of the color ramp indicates that darker shades represent longer durations.

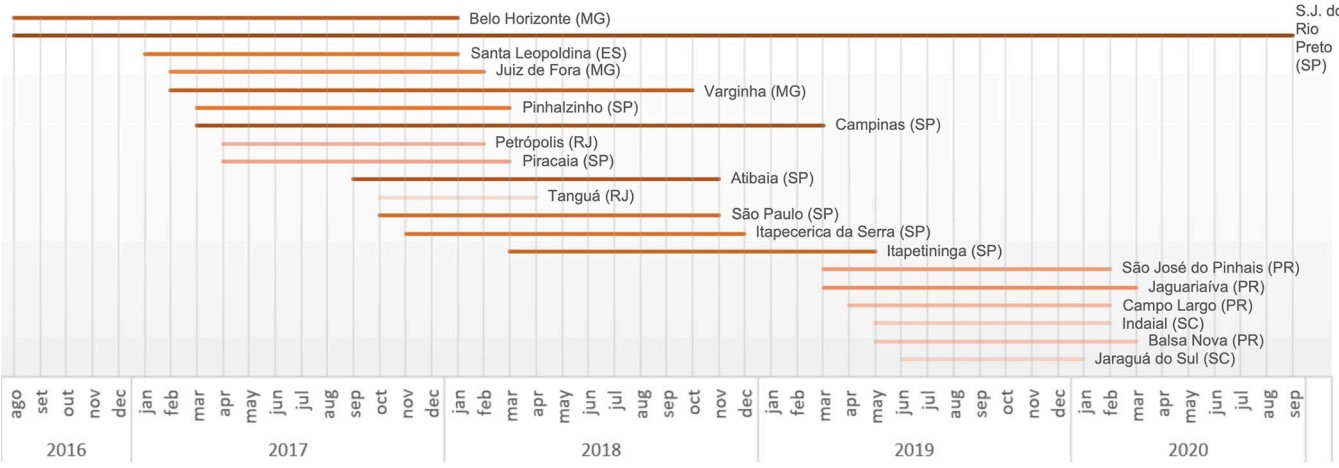

**Fig 8. Duration and distribution of human YF cases across municipalities.** The gradation of the color ramp indicates that darker shades represent longer durations.

## Discussion

The Atlantic Forest Biome, abundant with diverse NHPs and wild vectors, coupled with a densely populated human community, made the spread of YFV during the 2016–2020 outbreak particularly alarming. The period of 2017/2018 stands out as one of the most significant peaks in recorded occurrences of YF. Our results, which include analyses using regression models and Moran's I, suggest a correlation between YF human cases and epizootics, with both hosts being prevalent in a significant number of municipalities. In 2017 and 2018, respectively, 23.74% (61 out of 257) and 26.20% (71 out of 271) of municipalities experienced the simultaneous impact of both hosts.

Observing these trends, it becomes clear that the detection of NHP cases has led to increased surveillance of these species. During outbreaks, the reporting of NHP deaths by the Public Health System likely prompted intensified surveillance in areas with an increase in human cases. This suggests that there was a period during which NHP cases and human cases

**Table 2. Results of the association values and their significance derived from a spatio-temporal linear mixed-effects model analyzing epizootics and the prevalence of confirmed YF cases in humans.**

| NON-HUMAN-PRIMATES | Covariates | Value | Std. Error | DF | t-value | p-value | HUMANS | Covariates | Value | Std. Error | DF | t-value | p-value |
|---|---|---|---|---|---|---|---|---|---|---|---|---|---|
| | (Intercept) | -6.3919 | 0.1228 | 12314 | -52.0624 | <0.001 | | (Intercept) | -2.1932 | 0.0823 | 12312 | -26.6491 | <0.001 |
| | Forest Formation Area | 0.0325 | 0.0032 | 12314 | 10.0477 | <0.001 | | Epizootics | 2.7570 | 0.1209 | 12312 | 22.8095 | <0.001 |
| | Presence of Callithrix sp | 1.5116 | 0.1574 | 3077 | 9.6010 | <0.001 | | Temporary Crop | -0.0797 | 0.0084 | 12312 | -9.5050 | <0.001 |
| | Savanna and Grassland | -0.0027 | 0.0081 | 12314 | -0.3365 | 0.7365 | | Savanna and Grassland | -0.0332 | 0.0095 | 12312 | -3.5036 | <0.001 |
| | | | | | | | | Urban Area | -0.0491 | 0.0114 | 12312 | -4.3112 | 0.0012 |
| | R² cond | | | 0.459 | | | | R² cond | | | 0.263 | | |
| | R² marg | | | 0.106 | | | | R² marg | | | 0.263 | | |

Std. Error: Standard error; DF: degrees of freedom; R² cond = conditional R²; R² marg = marginal R².

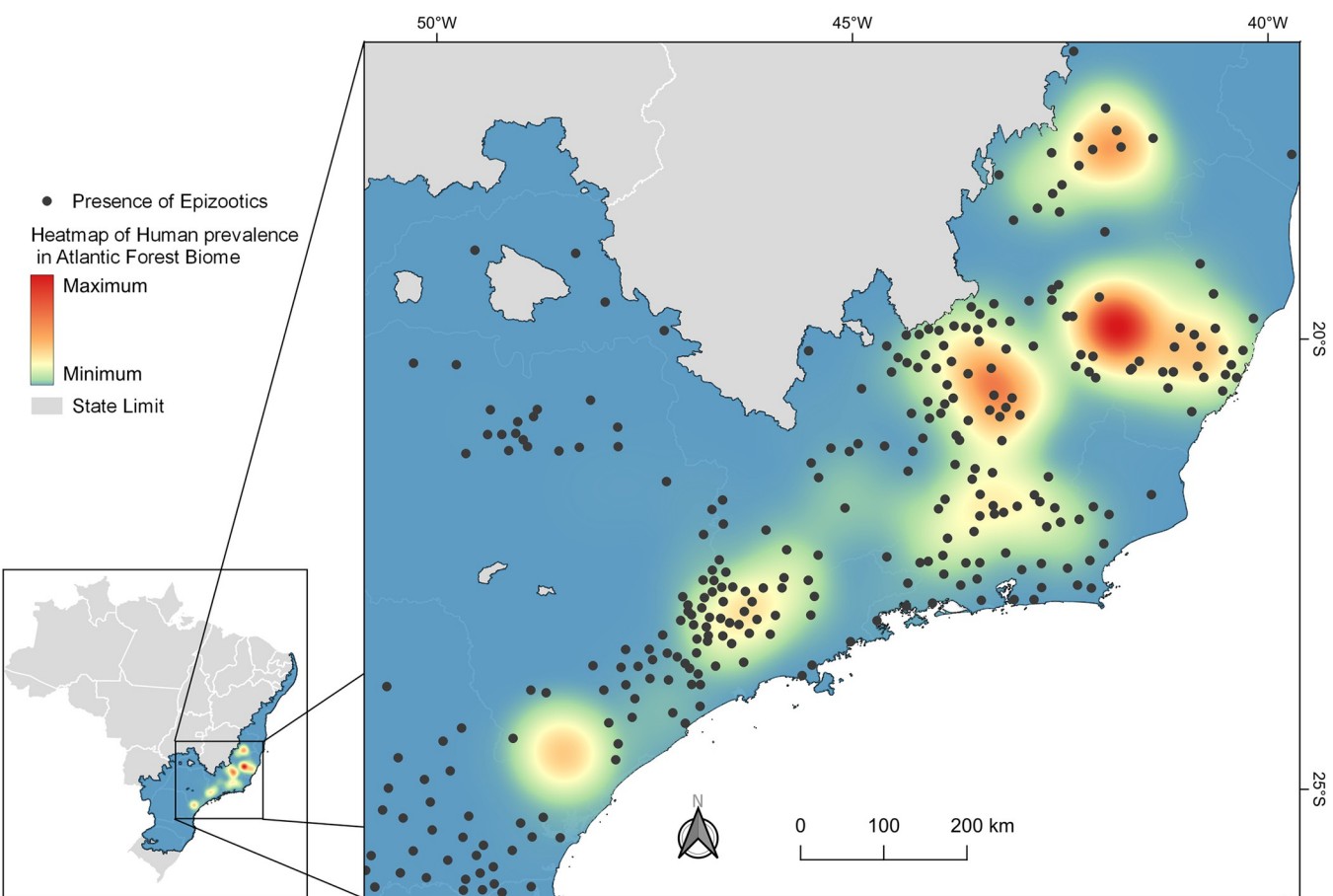

**Fig 9. Heatmap illustrating the prevalence of human cases and epizootics in the AFB from 2016 to 2020.**

**Table 3. Average, standard deviation (SD), and median values for significant variables in each model, stratified by municipalities with confirmed cases (MCC) and municipalities without confirmed cases (MNC).**

| NHPs | Variable | Association | | Average (± sd) | Median |
|---|---|---|---|---|---|
| | Forest Formation (%) | Direct | MCC | 32.90 ± 20.74 | 29.73 |
| | | | MNC | 22.03 ± 21.29 | 15.35 |
| | | | | | |
| | | | | | |
| HUMANS | Variable | Association | | Average (± sd) | Median |
| | Temporary Crop (%) | Inverse | MCC | 4.16 ± 9.66 | 0.10 |
| | | | MNC | 18.58 ± 22.40 | 8.06 |
| | Savannah and Grassland (%) | Inverse | MCC | 1.87 ± 6.49 | 0.00 |
| | | | MNC | 3.25 ± 8.99 | 0.00 |
| | Urban area (%) | Inverse | MCC | 5.59 ± 12.3 | 1.05 |
| | | | MNC | 3.20 ± 8.82 | 0.84 |

NHPs: Non-human primates

**Table 4. Moran's spatial analysis of NHP and human YF cases in AFB, 2016–2020.**

| | VARIABLE | ASSOCIATION | GMI | HH | LL | LH | HL |
|---|---|---|---|---|---|---|---|
| NHPs | Altitude | Direct | 0.163 | 8.48% | 42.08% | 31.89% | 1.85% |
| | Temporary Crop | Inverse | -0.152 | 1.35% | 46.60% | 36.68% | 9.76% |
| | Cfa Koppen index | Inverse | -0.123 | 2.21% | 48.05% | 36.31% | 8.64% |
| | Humidity Range | Inverse | -0.122 | 1.72% | 44.92% | 29.55% | 8.21% |
| | Forest Formation area | Direct | 0.119 | 7.26% | 40.83% | 31.53% | 1.62% |
| | Urban area | Direct | 0.116 | 4.78% | 55.64% | 12.83% | 3.59% |
| | Rainfall Range | Direct | 0.103 | 8.01% | 26.32% | 46.24% | 2.34% |
| | NHP Univariate | Direct | 0.194 | 9.83% | 49.34% | 23.02% | 0.30% |
| HUMANS | **VARIABLE** | **ASSOCIATION** | **GMI** | **HH** | **LL** | **LH** | **HL** |
| | Temporary Crop | Inverse | -0.213 | 0.13% | 44.95% | 37.90% | 11.41% |
| | Rainfall Range | Direct | 0.190 | 10.32% | 28.20% | 43.93% | 0.46% |
| | Altitude | Direct | 0.164 | 8.77% | 42.45% | 31.60% | 1.48% |
| | Cfa Koppen index | Inverse | -0.163 | 1.22% | 46.73% | 37.30% | 9.96% |
| | Humidity Range | Inverse | -0.149 | 1.19% | 44.39% | 30.08% | 8.74% |
| | Epizootics | Direct | 0.105 | 10.16% | 49.51% | 22.66% | 0.16% |
| | Human Univariate | Direct | 0.126 | 10.98% | 62.96% | 18.14% | 0.33% |

*GMI: Global Moran Index; HH: high-high clusters (high values of both variables in a high-value neighborhood); LL: low-low clusters (zero or low values of both variables in a low-value neighborhood); LH: low-high (low value of dependent variables in a high-value neighborhood of independent variables); HL: high-low spatial outlier (high value of dependent variables in a low-value neighborhood of independent variables).

were identified concurrently. Analyzing municipalities that reported only NHP or human cases during our study period indicates that in isolated epizootics, human vaccination efforts may have been effective [67]. Conversely, when only human cases were reported, inadequate epizootic surveillance or cases near political municipal boundaries may have resulted in NHP infections being reported in neighboring municipalities.

Additionally, this observed temporal dependency underscores the importance of surveillance in epizootics for developing strategies to reduce YF cases in humans. Following the implementation of these measures, there was a significant decrease in the number of YF cases (humans, NHP, and across both hosts) as well as in the number of affected municipalities in 2019. In 2020, almost no municipalities reported human cases (only 1), while 6 municipalities reported cases in both humans and animal hosts. However, a significantly higher number (65 municipalities) reported only epizootics.

Moreover, 10.90% of municipalities reported NHP cases for more than one year, compared to 13.33% of municipalities with human cases across all states in the Southeast Region. This region, which is ecologically conducive to the maintenance of YFV, demonstrates a persistence of the virus during both epidemic and interepidemic periods, as identified by Rezende et al. [68]. Abreu et al. [4] expand on this concept by stating that viral circulation can continue undetected for at least three consecutive transmission seasons. This persistence is attributed to the mosaic-like pattern of the Atlantic Forest, where isolated fragments remain unaffected. The situation is further compounded by the diversity and abundance of NHPs.

Notable examples of this persistence are seen in three municipalities where epizootics persisted for more than two years. In Belo Horizonte (Minas Gerais), not only were the NHPs from this city subjected to rigorous testing, but it is likely that those from neighboring cities were as well, which could explain the prolonged duration of the study [31]. São José do Rio Preto (SP) has been identified as a region at risk for YFV spillover, as mapped by Li et al. [52]. This area is in close proximity to São Paulo city and other densely populated municipalities.

According to Lacerda et al. [15], Atibaia (São Paulo) was the second-highest municipality in terms of autochthonous human YF cases in 2017 and 2018. Our results indicate that in 2018, Atibaia ranked third among all municipalities in AFB, with a total of 40 human cases.

However, it is important to note that these municipalities, being tourist zones, might have had lower vaccine coverage among residents. The vaccination strategy promoted by SES-SP may not have achieved adequate coverage, particularly as the efforts were primarily focused on the resident population. This approach likely overlooked tourists, who may have had lower vaccination rates since YF vaccination was not mandated in the area at the time. The section on the probable place of infection (PPI) includes data from both residents and tourists and serves as a method to mitigate biases from unmeasured factors like human mobility, which affect the spatial and temporal dynamics of YF cases. Additionally, regarding the persistence of human YF cases, the cities of Valença in Rio de Janeiro (RJ) and Juiz de Fora in Minas Gerais (MG) experienced a duration of 14 months between the first and last cases during the period from 2016 to 2020. These municipalities recorded positive YFV pools of *Haemagogus* spp., as documented by Abreu et al. [4]. Moreover, the abundance of this vector species is positively correlated with the likelihood of infection [33].

Analyzing the spread of YF in various hosts over time indicates a significant shift of cases toward the Southern Region, characterized by intermittent occurrences within a brief time-frame. This suggests that the outbreak likely originated from multiple initial sources, accompanied by rapid virus dispersal. According to current understanding, the YFV lineage linked to this dispersion had been circulating for at least two years prior to the outbreak. It likely traveled from the Midwest region to the Southeast region or to the Brazilian Cerrado biome [28, 68]. The epicenter of the epizootics of AFB began in Minas Gerais in 2016 and subsequently split into two trajectories: one spreading towards São Paulo (2016–2020) and Paraná (2019–2020), and the other extending to Espírito Santo, Rio de Janeiro, and Bahia between 2017 and 2019. This dispersion was more diffuse and geographically widespread at the onset of the outbreak compared to human cases. Given the patterns of the sylvatic cycle in YF outbreaks, it was expected that reports from NHPs and humans would overlap, since epizootics usually precede and ultimately coincide with human cases [3, 8, 23, 31]. Given that the same YFV lineage persisted from 2017 to 2018, and shared a common recent ancestor identified in São Paulo in 2016 [68], concurrence with Thoisy et al. [31] is justified concerning the apparent shortcomings in YF surveillance using sentinel NHPs. This suggests there was a delay in implementing effective control measures. Interestingly, although they persisted for up to two consecutive years, the viral sub-lineages responsible for the 2016–2019 outbreak in the southeast disappeared from the region, spreading to the southern part of the country [19, 21]. In 2021, the introduction of a new YFV strain in the southeast was traced back to a sub-lineage that had circulated in the Amazon in 2017 [11, 69]. It is noteworthy that Bahia, a state bordering the north of Espírito Santo and Minas Gerais, stood out as an exception in the dispersion of the virus. Despite the detection of epizootics in 21 municipalities (5.0% of 417), no human cases were reported. The affected municipalities are located in the following macroregions: East (52.4%; 11), Northeast (23.8%; 5), Central-East (19.0%; 4) and South (4.8%; 1). This success is attributed to effective epidemiological surveillance strategies [14]. Health authorities in Bahia proactively expanded YF vaccination efforts in municipalities in the extreme south in response to cases in nearby states in 2017 [70]. They also implemented vector control measures [14].

A detailed analysis of YF cases by state over the years shows that in 2017 and 2018, Minas Gerais, Espírito Santo, Rio de Janeiro, and São Paulo recorded a higher number of human cases compared to epizootics. This trend can likely be attributed to the peak of the outbreak in 2017–2018, which was characterized by inadequate surveillance strategies, high population density, a lack of YFV vaccine recommendations, and limited availability of the YF vaccine on

a large scale [4, 15, 16]. Since the implementation of vaccine coverage in 2017, Paraná and Santa Catarina have shown fewer human cases compared to epizootics in both 2019 and 2020 (Fig 5). Although infant vaccine coverage (IVC) did not show a significant association with any host model, it exhibited a significant and inverse correlation with urban areas ($\rho$ = -0.28; p < 0.05) in the human model.

Brazil began its yellow fever vaccine (YFVac) immunization campaigns in 1937 and included the vaccine in the public vaccination schedule for infants in endemic areas starting in 1991. The nationwide vaccination program was launched in 1998 but was later restricted to areas with high incidence rates in 2000 following reports of severe adverse effects from the mass vaccination effort [71]. Safety concerns, particularly heightened risks associated with the first vaccination and increasing with age, prompted authorities to recommend childhood immunization only in at-risk areas [72, 73]. The impact of this decision was clearly evident in São Paulo and Rio Grande do Sul during the 2008–2009 outbreak, where preventive mass vaccination could have been instrumental [8]. In 2013, the Brazilian Ministry of Health established a vaccination goal to achieve 100% coverage for the population up to one year of age, with booster doses administered every ten years [74]. As of 2021, Brazil achieved a vaccination coverage rate of 56.7%, with Santa Catarina at the forefront with 74.5%, closely followed by Minas Gerais at 73.8%. However, these figures are still below the recommended 80% threshold [75]. It is imperative to intensify efforts to reach higher vaccination rates, considering the essential role of vaccines in preventing the spread of viruses among both hosts. With the entire region now designated as an area recommended for vaccination, it is crucial that local health authorities strive to expand vaccination coverage, particularly in areas most susceptible to YFV transmission, thereby safeguarding a larger portion of the population. Conversely, structuring and maintaining epizootic surveillance teams is crucial for the early identification of new virus circulations among NHPs. This ensures that targeted vaccination efforts can be intensified in affected areas, thereby protecting individuals who may not yet have been vaccinated.

Altitude was identified as a variable that directly correlates with both hosts in Moran's analysis. However, despite its significance, extremely high altitudes are not conducive to the transmission of YF, as the virus cannot be transmitted at altitudes above 2,300 meters [76]. This environmental factor, which influences temperature gradients, affects mosquitoes, virus viability, and the distribution of NHPs [49]. In a study by Hamrick et al. [49], municipalities in America situated at altitudes between 318 and 784 meters were found to have a risk of YF presence six times higher than those located at altitudes above 1809 meters. While Almeida et al. [42] used a maximum entropy algorithm that considered elevation as a potentially predictive variable for yellow fever, our study found that 70% of municipalities with confirmed human cases were situated at altitudes ranging from 331 to 915 meters. The average altitude of municipalities with confirmed cases was 659.27 meters (SD = 287.09 m; median = 710 m), whereas those without confirmed cases had an average altitude of 513.62 meters (SD = 290.88 m; median = 492.00 m).

In the NHP regression model, the occurrence of *Callithrix* sp. was positively associated with the presence of epizootics. Despite both *Callithrix* sp. (35.1%) and *Alouatta* sp. (11.7%) being noted to be susceptible to YFV [7, 9, 77–79], our study suggests a positive association between epizootics and *Callithrix* sp. This genus is synanthropic [9], underscoring the importance of surveillance to mitigate the risk of re-urbanization. The moderate correlation ($\rho$ = 0.30; p<0.001) between these species suggests that their presence was not consistent across all municipalities. Silva et al. [17] suggest that the genetic makeup of NHP species in Brazil varies geographically, influencing their vulnerability to YFV. Behavioral and ecological factors unique to each NHP genus/species, influenced by their geographic distribution, further affect their susceptibilities [17].

In the NHP regression model, the area of forest formation showed a positive correlation with the occurrence of epizootics, which is consistent with the expected patterns due to the sylvatic transmission cycle of YFV in Brazil. According to a theoretical model that simulates various scenarios of fragmented land covers, along with vector and host dispersion, the average speed of virus transmission was found to be higher in landscapes with a high proportion of forest cover and a lower density of edges [54]. While not included directly in the human regression model, forest formation exhibited a significant inverse correlation ($\rho$ = -0.33; p<0.05) with temporary crops, suggesting a relationship that warrants further investigation. The variable of forest formation is closely associated with the rural characteristics of the disease in humans.

Analysis of the urban area within the human regression model indicated a negative correlation with human prevalence, consistent with expectations given the lack of spillover events in urban settings [80]. This trend is particularly notable in areas where fragments of the Atlantic Forest are adjacent to urban zones [4]. Forest fragments surrounded by urban areas serve as barriers to the spread of the YFV. This is attributed to high urban temperatures, the lack of forest canopy, and prevalent air pollution, which together create an inhospitable environment for the wild vectors of YFV [25]. However, there is still a risk that the urban cycle vector, *Ae. aegypti*, could encounter YFV, potentially leading to a resurgence of urban YF.

The model's finding of a negative association between urban areas and human cases of YF underscores the disease's predominantly rural nature, highlighting that the highest risks are faced by those living or working in regions where sylvatic transmission is prevalent [20]. Additionally, the higher prevalence observed in adult men aligns with previous research that indicates their increased risk. This may be attributed to the greater proportion of males in rural areas and their involvement in riskier activities, often accompanied by less attention to health precautions [2, 17]. This can be attributed to the higher prevalence of males in rural areas, approximately 80.87% [81], coupled with the widely held belief that adult men are more likely to engage in risky activities and are less attentive to their health [9, 82]. It is important to note that the yellow fever vaccine, which is administered beyond childhood, may not have been effectively distributed to rural adult men prior to and during the outbreak studied, as the prioritization of vaccination within municipalities only received widespread attention in 2020.

The relationship between temporary crops and the presence of infected hosts was found to be inverse, indicating that this type of land use may not facilitate the movement of vectors and, consequently, the spread of viruses in the AFB. As discussed in [25], agriculture could serve as a barrier to virus dispersal, potentially due to the use of herbicides and pesticides on cultivated lands, which may affect the density of vectors.

Savannah and grassland were also identified as negative predictors of human YF cases. In the AFB, this type of land cover occurs in ecotone areas, representing a gradual transition between the Cerrado and Caatinga biomes in Brazil (S3E Fig). This inverse response could be anticipated, given that such land coverage appears less conducive to vectors and NHPs, offering fewer food sources and less favorable climatic conditions for tree-hole breeding mosquitoes such as *Hg. janthinomys* and *Hg. leucocelaenus*, which are considered the primary vectors [4, 83]. The Cerrado biome, which serves as a "corridor" for the spread of YFV from the Amazon to AFB [11], had already been included in vaccination recommendations prior to the 2016–2020 outbreak [8]. This could also explain why the savannah was inversely related in our model. Further details about the typical vegetation in Brazil's Cerrado and Caatinga biomes indicate that the climate conditions are classified according to the Köppen system as Aw and BSh, respectively, with smaller areas categorized as Cfa [55, 58]. These classifications entail as: Aw) tropical with dry winter (annual rainfall <25mm, cold temperature ≥18˚C); BSh) Dry, Semi-arid with low latitude and altitude (annual rainfall ≥5mm, annual temperature ≥18˚C);

Cfa) Humid subtropical with oceanic climate without a dry season and a hot summer and with a high-temperature range (dry rainfall >40mm, hot temperature $\geq$22˚C, cold temperature $\geq$-3˚C to <18˚C). These climatic factors display characteristics that impede the reproduction and dispersal of *Haemagogus* vectors. These align with the factors identified in our model, which include Aw (dry winter), Cfa (high thermal range throughout the year), and BSh (dry period and low altitude). Notably, the mentioned altitude shows a direct association with both hosts in Moran's analysis.

While it is well-known that vectors are significantly influenced by climatic conditions, our model did not demonstrate a direct correlation between climate and the prevalence of infected hosts. This could be attributed to seasonal variations throughout the year, as our analysis considered the year as a time unit rather than a month. The municipal scale employed may overlook the microenvironments favored by vectors. Additionally, in some regions, epizootic events are directly linked to human YF cases, which leads to increased zoonotic surveillance and the potential for overreporting in specific areas. The detection of epizootic diseases introduces surveillance biases, especially in NHPs, which can affect regression analyses [24]. To reduce this bias, we employed Moran's analysis, taking into account neighboring reports to improve the reliability of case reporting.

Despite the comprehensive analysis and valuable insights provided by this study on yellow fever in the Atlantic Forest region, it is important to acknowledge certain limitations. The dependence on surveillance data, which is prone to underreporting and geographic biases, can affect the conclusions reached. Additionally, the use of a municipal scale may mask variations in microenvironments that influence disease dynamics. While regression analyses are informative and can demonstrate associations, they do not necessarily establish direct causation between variables. It should also be noted that climate patterns can vary throughout the year, potentially affecting the relationships among the factors studied. This study enhances our understanding of the interplay among environmental, urban, and epidemiological factors in the spread of yellow fever in the Atlantic Forest, thereby laying a groundwork for future research and informing public health strategies in the region. It is important to note that, although our model is based on a comprehensive database, variations may still occur due to gaps in knowledge about vector presence and viral lineages. Thus, it is essential to intensify monitoring of vector species in regions where cases have not yet been reported and to document human cases based on the genetic traits of the viruses. This approach will enhance our understanding of the dynamics of the yellow fever virus and contribute to the development of more effective control strategies.

Analyzing how various factors in our study interact with YF epizootics and human cases across different biomes offers a new viewpoint on the dynamics of the disease. This approach highlights the importance of environmental and climatic conditions rather than political or territorial boundaries. The presence of *Callithrix* is directly associated with epizootics, which in turn are linked to the prevalence of disease in humans. Epizootics are directly associated with forest formation. Conversely, human prevalence is inversely associated with urban areas, temporary crops, and savannah and grassland. Other factors show varying associations with the prevalence of each host, emphasizing spatial and temporal relationships, such as altitude, rainfall range, and humidity range. Enhancements in NHP surveillance, broader vaccine coverage, and improved vector control in agricultural activities can effectively mitigate the expansion of the disease.

## Supporting information

**S1 Table. Variables, sources, descriptions, and applications in each model of YF cases in municipalities within the Atlantic Forest biome of Brazil (2016–2020). —**$^*$ factors that were

not applied in the referenced model; D: dependent variable; I: Independent variable; t0: current year; t-1: previous year.
(PDF)

**S2 Table. Percentage and number of municipalities with confirmed YF cases by Biome, 2016–2020.**
(PDF)

**S3 Table. Correlations among independent variables show significant associations (p $<$ 0.05).**
(PDF)

**S4 Table. Repetition of confirmed yellow fever cases in municipalities within the Atlantic Forest biome, categorized by Brazilian state, 2016–2020.**
(PDF)

**S1 Fig. Area of cases and vaccine recommendation in Brazil.** A) Human confirmed cases of YF through 2016–2020, B) Infant Vaccine Coverage in 2015, C) Infant Vaccine Coverage at the beginning of AFB outbreak in 2016, D) Infant Vaccine Coverage during the outbreak in 2017 to 2020.
(TIF)

**S2 Fig.** QQ plot depicting the distribution of A) human prevalence; and B) log-transformed human prevalence for normality assessment.
(TIF)

**S3 Fig. Significant factors in NHP and human models by municipality.** A) Presence of *Callithrix* sp.; B) Percentage of forest cover in the AFB and in states containing at least one municipality within the AFB; C) Occurrence of epizootics; D) Proportion of land dedicated to temporary crops within the AFB; E) Percentage of savannah and grassland across Brazil; and F) Extent of urban areas within the AFB. Categories graduated by natural breaks.
(TIF)

**S4 Fig.** Univariate Local Moran's I for A) Epizootic Events; and B) Human Prevalence. "High-High" indicates a high intensity of the variable, while "Low-Low" signifies the absence of the variable.
(TIF)

**S5 Fig.** Bivariate Local Moran's I analysis of epizootics in relation to A) altitude; B) urban areas; C) forest formations; D) rainfall ranges; E) temporary crops; and F) Köppen Cfa index. High-High: high intensity of both variables; Low-Low: absence of both variables; Low-High: high intensity of one variable and absence of the other. The absence of epizootics and a high value of the independent variable; High-Low: The presence of epizootics and a zero or low value of the independent variable.
(TIF)

**S6 Fig.** Bivariate Local Moran's I analysis of human prevalence in relation to A) rainfall variability, B) altitude, C) epizootic events, D) temporary crops, E) Köppen Cfa index, and F) humidity variability. High-High: high intensity of both variables; Low-Low: absence of both variables; Low-High: high intensity of one variable and absence of the other. Low or zero human prevalence and a high value of the independent variable; High-Low: High human prevalence and negligible or zero values of the independent variable.
(TIF)

**S1 Appendix. Script of variables analysis and model development.**
(PDF)

## Acknowledgments

We are grateful for the Febre Amarela Project and their researchers and partners (https://www.febreamarelabr.com.br/), including Brazilian Ministry of health.

## Author Contributions

**Conceptualization:** Maíra G. Kersul, Filipe V. S. Abreu, Adriano Pinter, Anaiá da P. Sevá.

**Data curation:** Maíra G. Kersul, Miguel de S. Andrade, Anaiá da P. Sevá.

**Formal analysis:** Maíra G. Kersul, Anaiá da P. Sevá.

**Funding acquisition:** Filipe V. S. Abreu, Adriano Pinter, Fabrício S. Campos, Miguel de S. Andrade, Danilo S. Teixeira, Marco A. B. de Almeida, Paulo M. Roehe, Ana Claudia Franco, Aline A. S. Campos, George R. Albuquerque, Bergmann M. Ribeiro, Anaiá da P. Sevá.

**Investigation:** Maíra G. Kersul.

**Methodology:** Maíra G. Kersul, Filipe V. S. Abreu, Adriano Pinter, Fabrício S. Campos, Marco A. B. de Almeida, Paulo M. Roehe, Anaiá da P. Sevá.

**Project administration:** Anaiá da P. Sevá.

**Supervision:** Anaiá da P. Sevá.

**Writing – original draft:** Maíra G. Kersul, Anaiá da P. Sevá.

**Writing – review & editing:** Filipe V. S. Abreu, Fabrício S. Campos, Danilo S. Teixeira, Marco A. B. de Almeida, Paulo M. Roehe, Aline A. S. Campos.

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
