## [Decision Letter · Decision Letter 0]

5 Jun 2024

PONE-D-24-06374Exploring environmental and climate features associated with yellow fever across space and time in the Brazilian Atlantic Forest biomePLOS ONE

Dear Dr. Guimarães Kersul,

Thank you for submitting your manuscript to PLOS ONE. After careful consideration, we feel that it has merit but does not fully meet PLOS ONE’s publication criteria as it currently stands. Therefore, we invite you to submit a revised version of the manuscript that addresses the points raised during the review process.There are mainly clarification of the forest biome definition or side information you did not provided as the mosquitoes density.

We look forward to receiving your revised manuscript.

Kind regards,

Pierre Roques, Ph.D.

Academic Editor

PLOS ONE

Journal Requirements:

 [We are grateful to National Council of Scientific Development (Conselho Nacional de Desenvolvimento Científico e Tecnológico – CNPq, chamada CNPq/MS-SCTIE-Decit No 22/2019, process number 443215/2019-7) and to the Brazilian Office to Coordinate Improvement of Higher Education Personnel (CAPES) for the fellowship for Maíra G. Kersul from (process number 88887.604207/2021-00)].  

5. Please include captions for your Supporting Information files (Supporting Information S1 Appendix) at the end of your manuscript, and update any in-text citations to match accordingly. Please see our Supporting Information guidelines for more information: http://journals.plos.org/plosone/s/supporting-information. 

Additional Editor Comments:

The 3 reviewers find the work of great interest and quite strong and suggest some minor modifications and clarifications. Most of them should be included in the revision but some request specific answers to the reviewers.

Reviewers' comments:

Reviewer's Responses to Questions

**Comments to the Author**

1. Is the manuscript technically sound, and do the data support the conclusions?

Reviewer #1: Yes

Reviewer #2: Yes

Reviewer #3: Partly

2. Has the statistical analysis been performed appropriately and rigorously? 

Reviewer #1: Yes

Reviewer #2: Yes

Reviewer #3: I Don't Know

3. Have the authors made all data underlying the findings in their manuscript fully available?

Reviewer #1: Yes

Reviewer #2: Yes

Reviewer #3: No

4. Is the manuscript presented in an intelligible fashion and written in standard English?

Reviewer #1: Yes

Reviewer #2: Yes

Reviewer #3: Yes

5. Review Comments to the Author

Reviewer #1: Karsul et al. presents a manuscript in which they analyzed data collected from 2016 to 2020 that focused on the occurrence of Yellow Fever (YF) in humans and non-human primates (NHP). A generalized linear mixed regression model was employed to assess the spatial-temporal distribution of YF cases in human and NHP in the Atlantic Forest Biome. The analyses took into account several essential factors, including environmental characteristics, climate factors, vaccinations programs, and the occurrence of YF in at least 2 NHP species, (and perhaps to a much less extent the prevalence of known vector mosquito species in municipal regions). Based on extensive in silico analyses, the data convincingly showed that epizooitic can be correlated to natural forest formation and the presence of the NHP, Calithrix species, and that the epizootics coincided with human cases of YF. The data also revealed several important associations, for example, ~93% of all NHP YF cases occurred with the Atlantic Biome (AB), where these cases peaked in 2017; ~98% of Brazilian human YF cases occurred in 2018 in the AB, not surprising following epizootics; and lower number of cases where vaccination programs were implements. Though the results of the study are largely predictable, considering the occurrence of epizootics, forest expansion and management, and presence/prevalence of mosquito vector species, the extensive statistical analyses provide solid support for these observations.

General:

1. Statistical analyses were extensive and convincing

2. Conclusions are well supported.

3. The authors should provide specific references/citations for each of the in silico method used, including online links.

4. Could the authors in include reliable information on the prevalence of the mosquito vector species for humans and NHP in the larger areas surveyed? Are data available to demonstrate specific vector population increases and decreases that correlate to the prevalence of YF in both human and NHP?

5. Substantial editing of the manuscript is required. The Reviewer suggests the authors enlist the assistance and manuscript editing firm to make the appropriate changes.

One example of required editing in the first sentence of the ABSTRACT:

Line 36-39. "The Atlantic Forest is a biome (AFB) which hosts and favorable conditions for the occurrence

of the Yellow Fever virus (YFV) promoted by abundant vectors (Haemagogus and Sabethes

species) and several populations susceptible non-human primate (NHP) species (specially

Alouatta sp. and Callithrix sp.)."

Suggested change: "The Atlantic Forest biome (AFB) provides favorable conditions for the propagation of vector mosquitoes, including Haemagogus and Sabethes species, that disseminate the Yellow Fever virus to both human and non-human primates (NHP), particularly Alouatta sp. and Callithrix sp.)."

Reviewer #2: 1. The manuscript titled "Exploring environmental and climatic features associated with yellow fever across space and time in the Brazilian Atlantic Forest biome" presents a study that analyzes the spatiotemporal distribution of yellow fever (YF) cases in humans and non-human primates (NHP) in the Atlantic Forest biome of Brazil between 2016 and 2020. Data collection was based on official records of human cases and epizootics provided by the Brazilian Ministry of Health, encompassing data from Brazilian municipalities within the Atlantic Forest biome with confirmed cases, providing a robust basis for analysis. Although the analyzed data rely on epidemiological surveillance, subject to underreporting and biases in the widely collected information, the authors took these limitations into account in their discussion. Nonetheless, it would be unfeasible to control all variables given the magnitude of the research.

2. For data analysis, a Generalized Linear Mixed Model (GLMM) with a space-time framework was employed to assess the association between dependent variables (presence of epizootics and human prevalence) and a set of independent variables (such as environmental, climatic, vaccination coverage, and presence of NHP species). These models accounted for both fixed and random effects, controlling for time and space. Spatial correlation between municipalities was included in the analysis, conducted using the R software. Results were explored to understand variable behavior patterns in municipalities with and without confirmed disease cases. The authors appropriately opted for logarithmic transformation to normalize data regarding human prevalence. They were also meticulous in selecting independent variables, choosing those with significant correlation with the dependent variable to ensure validity and robustness of results. The use of GLMMs provides a robust approach to analyze the geographic distribution of diseases like yellow fever, controlling for fixed and random effects while incorporating spatial correlation structure. However, availability of more precise data regarding case geolocation could offer a more detailed analysis of spatial distribution patterns of the disease and facilitate better understanding of variable interactions, resulting in deeper insights into the determinants of yellow fever occurrence. While the current approach is suitable with available data, inclusion of more refined data could further strengthen conclusions and guide more precise interventions in combating the disease.

3.The authors provided the data in full in the form of maps, tables, and supplementary data.

4. The research is described clearly, with detailed explanations of the methods. The results are presented in a structured manner, using maps, tables, and supplementary data that facilitate understanding. The conclusions are well-supported by the data presented, making the manuscript suitable for the target audience.

The dynamics involved in the circulation of the yellow fever virus are complex and entail the interaction of environmental, climatic, and host factors as described in the study. The research attempts to address various factors but may oversimplify some relationships due to limitations in the data and methods used. It is important to consider that findings may be influenced by other unmeasured factors not accounted for in the analysis, such as the presence of wild and urban vector species, human mobility, and genetic variation of the virus. While the study provides valuable insights, the conclusions should be interpreted with caution.

Reviewer #3: Line 104- Maybe include references: da Silva et al, 2010: Hygeia 6(10):77 - 89, Jun/2010 and Couto-Lima et al, 2020: doi: 10.1590/0074-02760200218

Line 118- It is necessary to mention which criteria were used to delimit the Atlantic Forest Biome. Wouldn't official instruments be better, such as Federal Law 11,428/2006 and Decree 6,660/2008? It is possible that any change in this regard could alter the results.

Line 145- It is indeed opportune to use a population denominator to weight the cases; Wouldn't the rural population be better? Or maybe test both?

Line 149- In addition to presence/absence, in the same way as humans, it wouldn't be interesting to ponder; It is clear that there is no way to use population data on wild primates, but could it not be weighted by the area of the forest environment, which is, after all, the characteristic habitat of these animals? Or perhaps due to the total number of animals reported even without a confirmed laboratory diagnosis (at least there would be a dimension of the number in each municipality)?

Lines 151/152- I don't consider this criterion very clear; As I mentioned previously, a possible criterion is what delimits the Atlantic Forest by Decree; or even, studies such as Ribeiro et al., 2009-doi: 10.1016/j.biocon.2009.02.021 or Ribeiro et al., 2011-doi: 10.1007/978-3-642-20992-5_21

Line 177- Interesting: is there any study that led the authors to determine deforestation as an important variable in such a short period?

Line 195- I think that the variables related to the two types of climates are very generic; Why did the authors imagine it would be possible to reveal any association?

Lines 203- Why not the global vaccination rate?

“Descriptive analysis”: I suggest supplementary figures showing the area of vaccine recommendation before 2016, and also vaccination coverage by municipalities before and after the study period.

Line 285- In this description of the mathematical formula, the example of the years "2007, 2008,..." confuses the reader; better to put "2016, 2017,...2020".

Fig 9- As in figures 3 and 4, it is more enlightening to keep the state boundary lines superimposed on the color gradient of the variable. Why was a heat map drawn up for human cases and not for epizootics?

S2 Fig.- Why did they only show the presence of Callithrix and not Alouatta? What exactly would this presence of Callithrix be; would it be the records of deaths and not just those positive for YF? Why is there representation in the maps of forest formation % and Savanna and Grassland (%) beyond the Atlantic Forest Biome polygon while the other descriptors are circumscribed?

6. PLOS authors have the option to publish the peer review history of their article (what does this mean?). If published, this will include your full peer review and any attached files.

Reviewer #1: No

Reviewer #2: No

Reviewer #3: **Yes: **Luís Filipe Mucci

---

## [Author Response · Author response to Decision Letter 0]

25 Jul 2024

Response to reviewers

We would like to thank the reviewers for their thorough analysis and valuable suggestions for improving our manuscript entitled “Exploring environmental and climate features associated with yellow fever across space and time in the Brazilian Atlantic Forest biome”.

 In response to the comments provided, we have meticulously reviewed the manuscript and incorporated the suggested changes to address all concerns raised. We believe that we have made all necessary corrections and followed the reviewers' recommendations. These modifications have been integrated into the revised version of the manuscript. We are now submitting two versions: one with tracked changes (using Microsoft Word’s Tracker changes tool); and a clean version with the changes incorporated. Below, we address our responses to the reviewer’s comments one by one, highlighted in blue.

Reviewer #1: 

Karsul et al. presents a manuscript in which they analyzed data collected from 2016 to 2020 that focused on the occurrence of Yellow Fever (YF) in humans and non-human primates (NHP). A generalized linear mixed regression model was employed to assess the spatial-temporal distribution of YF cases in human and NHP in the Atlantic Forest Biome. The analyses took into account several essential factors, including environmental characteristics, climate factors, vaccinations programs, and the occurrence of YF in at least 2 NHP species, (and perhaps to a much less extent the prevalence of known vector mosquito species in municipal regions). Based on extensive in silico analyses, the data convincingly showed that epizooitic can be correlated to natural forest formation and the presence of the NHP, Calithrix species, and that the epizootics coincided with human cases of YF. The data also revealed several important associations, for example, ~93% of all NHP YF cases occurred with the Atlantic Biome (AB), where these cases peaked in 2017; ~98% of Brazilian human YF cases occurred in 2018 in the AB, not surprising following epizootics; and lower number of cases where vaccination programs were implements. Though the results of the study are largely predictable, considering the occurrence of epizootics, forest expansion and management, and presence/prevalence of mosquito vector species, the extensive statistical analyses provide solid support for these observations.

General:

1. Statistical analyses were extensive and convincing

2. Conclusions are well supported.

3. The authors should provide specific references/citations for each of the in silico method used, including online links.

 We would like to thank you for this suggestion. We have added the references we used to develop the model in line 269 of the manuscript. The references are as follows:

- F. Dormann C, M. McPherson J, B. Araújo M, Bivand R, Bolliger J, Carl G, et al. Methods to account for spatial autocorrelation in the analysis of species distributional data: A review. Ecography (Cop). 2007;30(5):609–28.

- Guo C, Du Y, Shen SQ, Lao XQ, Qian J, Ou CQ. Spatiotemporal analysis of tuberculosis incidence and its associated factors in mainland China. Epidemiol Infect. 2017;145(12):2510–9.

- Sevá A da P, Mao L, Galvis-Ovallos F, Oliveira KMM, Oliveira FBS, Albuquerque GR. Spatio-temporal distribution and contributing factors of tegumentary and visceral leishmaniasis: A comparative study in Bahia, Brazil. Spat Spatiotemporal Epidemiol. 2023;47(May 2021).

4. Could the authors in include reliable information on the prevalence of the mosquito vector species for humans and NHP in the larger areas surveyed? Are data available to demonstrate specific vector population increases and decreases that correlate to the prevalence of YF in both human and NHP? 

Thank you for addressing this topic. Unfortunately, despite the importance of vectors in YFV dissemination, the distribution data for mosquitoes is very scarce. There is no comprehensive survey of mosquito distribution in the Atlantic Forest, and even less so for the prevalence of viral infection among these populations.

One study (Abreu et al., 2022) demonstrated that an increase in the abundance of Haemagogus mosquitoes correlates with a higher likelihood of encountering infected mosquitoes. However, these data are geographically limited and do not reflect infection rates in humans or NHPs. This localized nature of the data limits its applicability to broader areas.

We have incorporated the information about the correlation between mosquito abundance and infection likelihood into our discussion to maintain the focus and coherence of the manuscript in line 585 of the manuscript, as it can be seen below:

“These municipalities registered positive YFV pools of Haemagogus spp., as documented by Abreu et al. (4). Moreover, the abundance of this vector species is positively correlated with the likelihood of infection (33).”

4. Abreu FVS de, Delatorre E, Santos AAC dos, Ferreira-de-Brito A, Castro MG de, Ribeiro IP, et al. Combination of surveillance tools reveals that Yellow Fever virus can remain in the same Atlantic Forest area at least for three transmission seasons. Mem Inst Oswaldo Cruz. 2019;114(1):1–10. 

33. Abreu FVS, de Andreazzi CS, Neves MSAS, Meneguete PS, Ribeiro MS, Dias CMG, de Albuquerque Motta M, Barcellos C, Romão AR, Magalhães MAFM, Lourenço-de-Oliveira R. Ecological and environmental factors affecting transmission of sylvatic yellow fever in the 2017-2019 outbreak in the Atlantic Forest, Brazil. Parasit Vectors. 2022 Jan 10;15(1):23. doi: 10.1186/s13071-021-05143-0.

5. Substantial editing of the manuscript is required. The Reviewer suggests the authors enlist the assistance and manuscript editing firm to make the appropriate changes.

One example of required editing in the first sentence of the ABSTRACT:

Line 36-39. "The Atlantic Forest is a biome (AFB) which hosts and favorable conditions for the occurrence

of the Yellow Fever virus (YFV) promoted by abundant vectors (Haemagogus and Sabethes

species) and several populations susceptible non-human primate (NHP) species (specially

Alouatta sp. and Callithrix sp.)."

Suggested change: "The Atlantic Forest biome (AFB) provides favorable conditions for the propagation of vector mosquitoes, including Haemagogus and Sabethes species, that disseminate the Yellow Fever virus to both human and non-human primates (NHP), particularly Alouatta sp. and Callithrix sp.)."

We appreciate the suggestion and have incorporated the sentence into the new version of the manuscript. Additionally, the manuscript was reviewed by a professional translator to ensure clarity and readability. We uploaded its certificate at the “attached files”.

Reviewer #2: 

1. The manuscript titled "Exploring environmental and climatic features associated with yellow fever across space and time in the Brazilian Atlantic Forest biome" presents a study that analyzes the spatiotemporal distribution of yellow fever (YF) cases in humans and non-human primates (NHP) in the Atlantic Forest biome of Brazil between 2016 and 2020. Data collection was based on official records of human cases and epizootics provided by the Brazilian Ministry of Health, encompassing data from Brazilian municipalities within the Atlantic Forest biome with confirmed cases, providing a robust basis for analysis. Although the analyzed data rely on epidemiological surveillance, subject to underreporting and biases in the widely collected information, the authors took these limitations into account in their discussion. Nonetheless, it would be unfeasible to control all variables given the magnitude of the research.

We appreciate the reviewer’s comments and have made the necessary revisions to the manuscript to address each of their suggestions comprehensively and scientifically. We thank the reviewers for their valuable feedback, which has significantly enhanced the quality of our publication.

2. For data analysis, a Generalized Linear Mixed Model (GLMM) with a space-time framework was employed to assess the association between dependent variables (presence of epizootics and human prevalence) and a set of independent variables (such as environmental, climatic, vaccination coverage, and presence of NHP species). These models accounted for both fixed and random effects, controlling for time and space. Spatial correlation between municipalities was included in the analysis, conducted using the R software. Results were explored to understand variable behavior patterns in municipalities with and without confirmed disease cases. The authors appropriately opted for logarithmic transformation to normalize data regarding human prevalence. They were also meticulous in selecting independent variables, choosing those with significant correlation with the dependent variable to ensure validity and robustness of results. The use of GLMMs provides a robust approach to analyze the geographic distribution of diseases like yellow fever, controlling for fixed and random effects while incorporating spatial correlation structure. However, availability of more precise data regarding case geolocation could offer a more detailed analysis of spatial distribution patterns of the disease and facilitate better understanding of variable interactions, resulting in deeper insights into the determinants of yellow fever occurrence. While the current approach is suitable with available data, inclusion of more refined data could further strengthen conclusions and guide more precise interventions in combating the disease.

Dear reviewer, we appreciate your thorough examination of our model. We acknowledge that geolocating both human and NHP cases could enhance the depth of our analysis. Regrettably, we did not have access to such precise information within the scope of our study. Typically, official databases do not furnish such specifics. Hence, we chose to utilize municipal-level data to address this lack of information. Additionally, this decision was influenced by the vast spatial extent of our study area, the Atlantic Forest biome, which encompasses 3,079 municipalities, representing the smallest political and administrative divisions.

3. The authors provided the data in full in the form of maps, tables, and supplementary data.

We appreciate and thank the reviewer.

4. The research is described clearly, with detailed explanations of the methods. The results are presented in a structured manner, using maps, tables, and supplementary data that facilitate understanding. The conclusions are well-supported by the data presented, making the manuscript suitable for the target audience.

The dynamics involved in the circulation of the yellow fever virus are complex and entail the interaction of environmental, climatic, and host factors as described in the study. The research attempts to address various factors but may oversimplify some relationships due to limitations in the data and methods used. It is important to consider that findings may be influenced by other unmeasured factors not accounted for in the analysis, such as the presence of wild and urban vector species, human mobility, and genetic variation of the virus. While the study provides valuable insights, the conclusions should be interpreted with caution.

We appreciate your feedback and insights on this issue. We have acknowledged the potential biases and limitations of our model in the discussion section (line 738) of the manuscript, where we address unmeasured factors such as the presence of wild and urban vector species, human mobility, and genetic variation of the virus. Specifically, regarding human mobility, we accounted for the Probable Place of Infection (PPI) to minimize potential data recording errors. Further elaboration on these points can be found in the "Discussion" section (line 575 of the manuscript), as outlined below:

“ However, it is important to note that these municipalities, being tourist zones, might have had lower vaccine coverage among residents. The vaccination strategy promoted by SES-SP may not have achieved adequate coverage, particularly as the efforts were primarily focused on the resident population. This approach likely overlooked tourists, who may have had lower vaccination rates since YF vaccination was not mandated in the area at the time. The section on the probable place of infection (PPI) includes data from both residents and tourists and serves as a method to mitigate biases from unmeasured factors like human mobility, which affect the spatial and temporal dynamics of YF cases. ”

Reviewer #3: 

Q1. Line 104- Maybe include references: da Silva et al, 2010: Hygeia 6(10):77 - 89, Jun/2010 and Couto-Lima et al, 2020: doi: 10.1590/0074-02760200218

We appreciate the reviewer’s guidance on this matter. We have inserted the necessary information at line 75 of the manuscript.

Q2. Line 118- It is necessary to mention which criteria were used to delimit the Atlantic Forest Biome. Wouldn't official instruments be better, such as Federal Law 11,428/2006 and Decree 6,660/2008? It is possible that any change in this regard could alter the results.

We appreciate the reviewer’s guidance on this matter and acknowledge the importance of citing authoritative source. For our purposes, the IBGE’s criteria are more suitable, as their descriptions are more up to date (2019), and we utilized their tools to delineate the study area. Therefore, we have inserted the citation of IBGE study at line 92, along with the following explanation:

“Bomes are characterized by features as vegetation, physical conditions, and biological diversity. Their boundaries were determined using a 2019 shapefile from IBGE (43). The biome of each municipality was determined by using both their shapefiles and the intersection function in QGIS.”

43. Instituto Brasileiro de Geografia e Estatística - IBGE. Biomas e Sistema Costeiro-Marinho do Brasil: compatível com a escala 1:250 000 [Internet]. Coordenação de Recursos Naturais e Estudos Ambientais, editor. Vol. 45, Série Relatórios metodológicos v. 45. Rio de Janeiro: IBGE; 2019. 168 p. Available from: https://biblioteca.ibge.gov.br/index.php/biblioteca-catalogo?view=detalhes&id=2101676

Q3. Line 145- It is indeed opportune to use a population denominator to weight the cases; Wouldn't the rural population be better? Or maybe test both?

We appreciate the reviewer’s suggestion on this matter. We agree that considering the rural population is important. We accounted for the rural population at a specific stage in our analysis. However, all cases have PPI (Probable Place of Infection) in rural or wild areas. Therefore, it is not accurate to refer to urban areas. The justification needs to be adjusted: when we discuss PPI, we do not always know if the person lived in an urban area and got infected in a rural/wild area or if they were residents of the rural/wild area and got infected there. Hence, it is essential to include the entire population. We have included these aspects in the “Methods”, section 2.1 (line 123). As it can be seen below:

“The prevalence of YF in humans was determined by calculating the number of cases per 10,000 inhabitants in each municipality annually, based on the total population figures from the 2010 census conducted by the Brazilian Institute of Geography and Statistics (IBGE). This entire population was considered, rather than focusing solely on rural areas, because individuals can become infected in rural or wild areas and still be urban residents.” 

Q4. Line 149- In addition to presence/absence, in the same way as humans, it wouldn't be interesting to ponder; It is clear that there is no way to use population data on wild primates, but could it not be weighted by the area of the forest environment, which is, after all, the characteristic habitat of these animals? Or perhaps due to the total number of animals reported even without a confirmed laboratory diagnosis (at least there would be a dimension of the number in each municipality)?

We appreciate your thoughtful commentary. However, for NHPs, there are significant challenges with underreported deaths, which makes it difficult to accurately determine their true population for proportional analysis. Consequently, we believe that identifying at least one case of NHPs per municipality is sufficient, particularly bec

---

## [Editor Report · Decision Letter 1]

26 Jul 2024

Exploring environmental and climate features associated with yellow fever across space and time in the Brazilian Atlantic Forest biome

PONE-D-24-06374R1

Dear Dr. Guimarães Kersul,

We’re pleased to inform you that your manuscript has been judged scientifically suitable for publication and will be formally accepted for publication once it meets all outstanding technical requirements.

Kind regards,

Pierre Roques, Ph.D.

Academic Editor

PLOS ONE

Additional Editor Comments (optional):

Thank you for these well detailled and complete answers to the reviewers
---

## [Editor Report · Acceptance letter]

13 Aug 2024

PONE-D-24-06374R1 

PLOS ONE

Dear Dr. Kersul, 

I'm pleased to inform you that your manuscript has been deemed suitable for publication in PLOS ONE. Congratulations! Your manuscript is now being handed over to our production team.

Kind regards, 

on behalf of

Dr. Pierre Roques 

Academic Editor

PLOS ONE